# Accelerating Transformer Training: Architectural Symmetry, Positional Encoding, and Teleportation

## Abstract

As neural architectures continue to grow in complexity and scale, the development of advanced optimization techniques has become increasingly important. Teleportation has recently emerged as a principled approach for accelerating the convergence of gradient descent-based algorithms by traversing loss-invariant level sets to identify parameterizations with favorable geometric properties. Although prior teleportation methods have achieved notable success in feedforward and convolutional networks, extending these techniques to Transformer architectures presents unique challenges. In particular, existing approaches typically assume the symmetry structure of vanilla attention, overlooking the critical role of positional encodings, which fundamentally reshape architectural symmetries and render earlier analyses inapplicable. To address this gap, we present a systematic study of teleportation in Transformer-based models. We first characterize how the architectural symmetry of multihead attention is modified under two widely used positional encoding schemes–sinusoidal and rotary–and provide a comprehensive description of the resulting symmetry groups. Guided by these insights, we introduce a teleportation framework tailored to Transformers and evaluate its effectiveness across diverse configurations, datasets, and modalities. Our results demonstrate the versatility of teleportation, elucidate the interplay between positional encoding and architectural symmetry in Transformer optimization, and establish a foundation for the principled development of teleportation algorithms that fully exploit the symmetry structure of Transformer architectures.

## 1 Introduction

Training modern deep learning models, particularly large-scale architectures such as Transformers, is highly resource-intensive, requiring extensive computation and energy. As models and datasets grow, accelerating optimization has become a central research challenge with direct implications for feasibility and scalability. To address this challenge, a number of research directions have sought to improve training speed and stability. Early work focused on optimization algorithms such as momentum-based methods (Sutskever et al., 2013), Adam (Adam et al., 2014), and its variants like AdamW (Loshchilov & Hutter, 2017). Beyond refining the optimization algorithm itself, subsequent research has explored more fundamental changes to the training dynamics, such as directly manipulating the parameter space to escape challenging geometries.

**Teleportation.** Recently, teleportation has been proposed as a principled approach to accelerate optimization by exploiting architectural symmetries that reparameterize neural networks without changing their functional capacity (Armenta & Jodoin, 2021; Saul, 2023). Unlike conventional gradient-based methods that advance through incremental updates, teleportation directly moves parameters to functionally equivalent states, thereby improving convergence efficiency (Zhao et al., 2022a; Mishkin et al., 2023) while also facilitating broader exploration of the loss landscape in contexts such as generalization (Zhao et al., 2022a) and privacy (Maheri et al., 2025).

**Functional Equivalence.** The effectiveness of teleportation fundamentally relies on functional equivalence, which asserts that distinct parameter configurations can realize the same network function (Armenta & Jodoin, 2021; Saul, 2023). This perspective explains why teleportation preserves

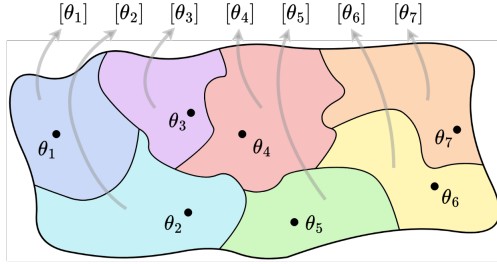 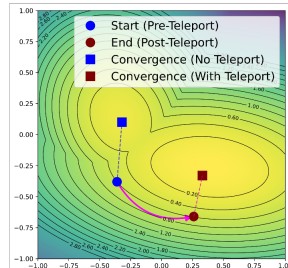

Figure 1: *(Left)* Partition of the parameter space into functional equivalence classes, as stated in Section 2. *(Right)* Illustration of teleportation in the optimization process: a point is mapped to another parameterization that realizes the same function but alters the optimization dynamics and trajectory, potentially leading the optimizer to a different local minimum.

expressivity and identifies the admissible directions along which parameters may vary without altering the underlying function. This principle has been applied across diverse architectures, including multilayer perceptrons (Zhao et al., 2022a; Mishkin et al., 2024; Zhao et al., 2023), convolutional networks (Armenta et al., 2023; Maheri et al., 2025), recurrent models such as LSTMs in reinforcement learning (Zamir et al., 2025), and continual learning frameworks with low-rank adaptation (Zhou et al., 2025). In contrast, only a few studies have investigated teleportation for Transformers, and these remain confined to small-scale settings such as MNIST, time-series forecasting, and Penn Treebank (Wu et al., 2025), leaving large-scale training largely unexplored. A key reason is that functional equivalence in attention-based models has been scarcely studied, with existing analyses limited to vanilla multihead attention (Tran et al., 2025; Knyazev et al., 2024).

**Attention and Positional Encoding.** The effectiveness of token teleportation in Transformers crucially depends on positional encoding, since self-attention is permutation invariant and requires explicit order information (Vaswani et al., 2017). Early approaches adopted Absolute Positional Encodings (APEs), either sinusoidal or learnable, which became standard in BERT (Devlin et al., 2019), GPT-2 (Radford et al., 2019), and ViT (Dosovitskiy et al., 2020). However, APEs generalize poorly to longer sequences (Press et al., 2021; Dai et al., 2019). To overcome this, Relative Positional Encodings (RPEs) (Shaw et al., 2018) were introduced, encoding pairwise distances directly into attention and yielding improved robustness in Transformer-XL, T5, and DeBERTa (Dai et al., 2019; Raffel et al., 2020; He et al., 2020). More recently, Rotary Positional Encoding (RoPE) (Su et al., 2024) extended this principle by embedding relative information through rotational transformations of query–key vectors, enabling translation equivariance and superior extrapolation. Its strong empirical performance has made RoPE a central component in modern large-scale models (Touvron et al., 2023; Chowdhery et al., 2023; Bai et al., 2025; Yang et al., 2025).

**Contributions.** Motivated by this line of work, we study the functional equivalence of Multihead Attention with positional encoding (PE), examining how it alters the symmetry structure of attention and its implications for teleportation training. The paper is organized as follows:

1. In Section 2, we examine the parameter space of a parameterized function, characterize its associated symmetry group, and introduce the formal notion of maximality within symmetry groups, establishing a direct connection to Functional Equivalence. We then compare with the finding on symmetry of vanilla attention in literature.

2. In Section 3, we analyze how positional encodings alter the internal structure of attention. We focus primarily on the most widely used encodings, Absolute PE and Relative PE. In particular, we study sinusoidal PE as a representative of APE and rotary PE as a representative of RPE, and show why results from the vanilla case do not extend directly to these settings. We then present our finding that fully characterizes the symmetry of attention with widely used PE.

3. In Section 4, we introduce our teleportation method based on sampling minimal perturbations along current optimization directions. This approach improves stability of teleportation steps while significantly reducing computational overhead compared to Hessian-based methods.

4. In Section 5, we report experimental results showing that our algorithm accelerates convergence, improves performance, and enhances generalization. We also present ablation studies that identify effective teleportation configurations across datasets of different scales.

A table of notation, theoretical foundations, and experimental details are provided in the Appendix.

## 2 FUNCTIONAL EQUIVALENCE AND MAXIMAL SYMMETRY GROUP

In this section, we formalize the parameter space of a parameterized function and its associated symmetry groups, culminating in the definition of maximal symmetry groups, which provide a principled link to Functional Equivalence (FE). We then specialize to Multihead Attention, examining how this notion of maximality aligns with prior analyses.

### 2.1 PARAMETER SPACE, SYMMETRY GROUP, AND ITS MAXIMALITY

**Parameter space.** Let $f(\cdot; \theta)$ be a function parameterized by $\theta \in \Theta = \mathbb{R}^{\dim}$. The set $\Theta$ is referred to as the *parameter space* (or *weight space*) of $f$. Assume a group $G$ acts on $\Theta$. For each $\theta \in \Theta$, define the set of parameter vectors yielding functionally equivalent models:

$$[\theta] := \left\{ \bar{\theta} \in \Theta \mid f(\cdot; \bar{\theta}) = f(\cdot; \theta) \right\} \subseteq \Theta. \tag{1}$$

The parameter space serves as a surrogate for the underlying function class, and the mapping $\theta \mapsto f(\cdot; \theta)$ is non-injective, since distinct parameter configurations may correspond to identical behaviors. This is illustrated in Figure 1. FE is therefore concerned with characterizing the sets $[\theta]$. As explicit enumeration is infeasible, a principled strategy is to interpret these equivalence classes as orbits under a group action on $\Theta$, leading naturally to the notion of the *symmetry group* of $f$.

**Symmetry group.** Let a group $G$ act on $\Theta$. For $\theta \in \Theta$, the $G$-orbit of $\theta$ is defined as $G\theta := \{g\theta \mid g \in G\} \subseteq \Theta$. We now state the following definition.

**Definition 2.1** (Symmetry Group). A group $G$ is called a *symmetry group* of the function $f$ if $G\theta \subseteq [\theta]$ for all $\theta \in \Theta$. Equivalently, for every $g \in G$ and $\theta \in \Theta$, one has $f(\cdot; g\theta) = f(\cdot; \theta)$.

The phrase "a symmetry group" reflects that multiple such groups may exist. In particular, any subgroup of a symmetry group is itself a symmetry group. Our objective is to represent the equivalence classes $[\theta]$ in terms of $G$-orbits. To develop intuition, we begin with two preliminary observations.

*First observation.* Consider the function $f(\cdot; a, b) \colon \mathbb{R} \to \mathbb{R}$ defined by $x \mapsto abx$, parameterized by $\theta = (a, b) \in \Theta = \mathbb{R}^2$. It is immediate that $(a, b)$ and $(\bar{a}, \bar{b})$ yield the same function if and only if $ab = \bar{a}\bar{b}$. This naturally suggests a group action: let $\mathbb{R}^\times$ denote the multiplicative group of nonzero real numbers, and define the action of $c \in \mathbb{R}^\times$ on $(a, b) \in \mathbb{R}^2$ by $c \cdot (a, b) := (ac, c^{-1}b)$. It is straightforward to verify that $\mathbb{R}^\times$ is a symmetry group of $f$. However, it does not fully capture the equivalence classes. Indeed, for $(a, b) \in \mathbb{R}^2$ with $ab \neq 0$, one has

$$[(a, b)] = \{(\bar{a}, \bar{b}) \in \mathbb{R}^2 \mid ab = \bar{a}\bar{b}\} = \{(ac, c^{-1}b) \mid c \in \mathbb{R}^\times\} = \mathbb{R}^\times(a, b). \tag{2}$$

In contrast, for $(a, b) \in \mathbb{R}^2$ with $ab = 0$, one has $[(a, b)] = \mathbb{R}^\times(1, 0) \sqcup \mathbb{R}^\times(0, 1) \sqcup \mathbb{R}^\times(0, 0)$. Hence, $\mathbb{R}^\times$ provides an almost complete description of the functional partition, but does not account for the degenerate subset $\{(a, b) \in \mathbb{R}^2 : ab = 0\}$. It is difficult to identify a larger natural group whose action extends to cover these exceptional cases.

*Second observation.* From classical group theory, any partition of a set can be realized as the orbit decomposition of a suitable group action. Hence, there always exists a group $G$ with an action on $\Theta$ such that its orbits coincide with the functional partition. Nevertheless, constructing such a group generally requires explicit transformations, which are often intractable and impractical. In parameterized models, where $\Theta$ is a finite-dimensional real vector space, it is natural to restrict attention to group actions induced by standard operations such as addition, multiplication, or permutation.

These two observations highlight a trade-off: the *tractability* of the group action versus the *expressive capacity* of the functional partition. This motivates the notion of maximal symmetry groups.

**Maximal symmetry group.** We now introduce the notion of a maximal symmetry group.

**Definition 2.2** (Maximal symmetry group). (informal) For generic parameters, the symmetry group $G$ fully captures functional equivalence, up to a sufficiently small exceptional set.

In other words, let $\varepsilon$ denote a sufficiently small subset of $\Theta$, and consider the restricted domain $\Theta \setminus \varepsilon$. The group action of $G$ on $\Theta$ naturally restricts to $\Theta \setminus \varepsilon$. Then, for all $\theta, \bar{\theta} \in \Theta \setminus \varepsilon$ such that

$f(\cdot; \theta) = f(\cdot; \bar{\theta})$, there exists $g \in G$ with $\bar{\theta} = g\theta$. Hence, although there may exist parameters in $\Theta$ for which $G$ does not capture FE, this exceptional set is negligible, and $G$ may still be regarded as characterizing FE on $\Theta$. The subset $\varepsilon$ is typically taken to be the zero set of finitely many nonzero polynomials, i.e., a proper real algebraic variety, in line with prior work on FE in neural architectures (Hecht-Nielsen, 1990; Fefferman & Markel, 1993; Bui Thi Mai & Lampert, 2020).

**Definition 2.3** (Maximal symmetry group). A symmetry group $G$ is called *maximal* if there exists a proper real algebraic variety $\varepsilon \subsetneq \Theta$ such that, for all $\theta, \bar{\theta} \in \Theta \setminus \varepsilon$, whenever $f(\cdot; \theta) = f(\cdot; \bar{\theta})$, there exists $g \in G$ with $\bar{\theta} = g\theta$.

**Remark 2.4.** In the earlier example of $f(\cdot; a, b)$, let $\varepsilon = \{(a, b) \in \mathbb{R}^2 : ab = 0\}$. Here $\varepsilon$ forms a proper real algebraic variety, and the group $\mathbb{R}^\times$ serves as a maximal symmetry group of $f$.

## 2.2 THE CASE OF MULTIHEAD ATTENTION

**Parameter space.** Let $d$ denote the token dimension, $L$ the sequence length, and $h$ the number of heads, where all are positive integers. Define the space of token sequences as $\mathcal{S} := \sqcup_{L=1}^\infty \mathbb{R}^{L \times d}$. For a fixed head dimension $d_h$, let $W_i^Q, W_i^K, W_i^V, W_i^O \in \mathbb{R}^{d \times d_h}$ for each $i \in [h]$, and set $\theta = (W_i^Q, W_i^K, W_i^V, W_i^O)_{i=1}^h$. Given an input sequence $\mathbf{x} = (x_1, \dots, x_L)^\top \in \mathbb{R}^{L \times d} \subset \mathcal{S}$, the Multihead Attention (MHA) mechanism with $h$ heads is defined by

$$\mathrm{MHA}\,(\mathbf{x}\colon \theta) = \sum_{i=1}^h \mathrm{softmax}\left(\left(\mathbf{x}W_i^Q\right)\left(\mathbf{x}W_i^K\right)^\top\right) \cdot \left(\mathbf{x}W_i^V\right)\left(W_i^O\right)^\top. \tag{3}$$

Here, the softmax operator is applied row-wise to the similarity matrix $(\mathbf{x}W_i^Q)(\mathbf{x}W_i^K)^\top \in \mathbb{R}^{L \times L}$, producing the attention for $\mathbf{x}$. Each row forms a probability distribution that determines the relative influence of all input tokens on a given output token. In practice, the head dimension is set to $d_h = d/h$. The parameter space of the MultiHead map is then $\Theta := \left(\mathbb{R}^{d \times d_h}\right)^{4h}$.

**Maximal symmetry group.** Define the following group $G_{\mathrm{Att}} := S_h \times (\mathrm{GL}(d_h) \times \mathrm{GL}(d_h))^h$. This group is exactly the direct product of the permutation group $S_h$ with $h$ copies of $\mathrm{GL}(d_h) \times \mathrm{GL}(d_h)$. Each element $g \in G_{\mathrm{Att}}$ can be written as $g := (\sigma, (U_i, V_i)_{i=1}^h)$, where $\sigma \in S_h$ and $U_i, V_i \in \mathrm{GL}(d_h)$. The group $G_{\mathrm{Att}}$ acts naturally on the parameter space $\Theta$ as follows:

$$g\theta := \left(W_{\sigma(i)}^Q \cdot U_i^\top; W_{\sigma(i)}^K \cdot U_i^{-1}; W_{\sigma(i)}^V \cdot V_i^\top; W_{\sigma(i)}^O \cdot V_i^{-1}\right)_{i=1}^h. \tag{4}$$

It is evident that $G$ serves as a symmetry group of the MHA map. The reasoning is as follows: the general linear action cancels within the matrix multiplications, while the permutation action induced by $\sigma$ commutes with addition. Furthermore, $G$ is maximal, as formalized in the following result.

**Theorem 2.5** (See Tran et al. (2025)). *Consider two* MHA *maps with $h$ heads, parameterized by* $\theta = (W_i^Q, W_i^K, W_i^V, W_i^O)_{i=1}^h$ *and* $\bar{\theta} = (\bar{W}_i^Q, \bar{W}_i^K, \bar{W}_i^V, \bar{W}_i^O)_{i=1}^h$ *in $\Theta$, respectively. Assume that*

1. *All matrices $W_i^Q, W_i^K, W_i^V, W_i^O$ and $\bar{W}_i^Q, \bar{W}_i^K, \bar{W}_i^V, \bar{W}_i^O$, for all feasible $i$, are of rank $d_h$.*

2. *From $\theta$, the matrices $\{W_i^Q(W_i^K)^\top\}_{i=1}^h$ are pairwise distinct. The same condition holds for $\bar{\theta}$.*

*If the two* MHA *maps are identical, there exists $g \in G$ such that $\bar{\theta} = g\theta$.*

**Remark 2.6.** Note that the conditions on $\theta$ and $\bar{\theta}$ in Theorem 2.5 can both be expressed as the vanishing of finitely many nonzero polynomials. This corresponds precisely to the real algebraic variety $\varepsilon$ introduced in Definition 2.3 of maximal symmetry groups.

## 3 ON THE EFFECT OF POSITIONAL ENCODING ON SYMMETRY GROUPS

Our investigation examines how positional encodings (PEs) alter the structure of attention. In particular, we focus on *sinusoidal encoding* and *rotary encoding*, which serve as canonical examples of absolute and relative positional encoding approaches.

### 3.1 THE SETTING OF ABSOLUTE POSITIONAL ENCODING

**Sinusoidal Encoding.** Within Absolute PEs, positional information is encoded through a sequence of vectors $\mathbf{p} = \{p_i\}_{i=1}^\infty \subset \mathbb{R}^d$. For the *sinusoidal encoding* proposed in the Transformer architecture (Vaswani et al., 2017), the entries of each $p_m \in \mathbb{R}^d$ are specified as

$$p_{m,2k} = \sin\left(\frac{m}{10000^{2k/d}}\right), \text{ and } p_{m,2k+1} = \cos\left(\frac{m}{10000^{2k/d}}\right), \tag{5}$$

for $0 \le k < d/2$. For an input sequence $\mathbf{x} \in \mathcal{S}$ of length $L$, i.e., $\mathbf{x} = (x_1, \ldots, x_L)^\top \in \mathbb{R}^{L \times d}$, the positional encoding is applied additively, that is $\mathbf{x} + \mathbf{p} = (x_1 + p_1, \ldots, x_L + p_L)^\top$ (this is an abuse of notation), which is then supplied as input to the multihead attention, yielding $\text{MHA}_{\text{APE}}(\mathbf{x}: \theta) = \text{MHA}(\mathbf{x} + \mathbf{p}: \theta)$. Under this formulation, PE *does not alter the internal mechanism* of the MHA map in Equation (3); rather, it simply translates the inputs. The mapping $\mathbf{x} \mapsto \mathbf{x} + \mathbf{p}$ is bijective on $\mathcal{S}$. As a result, incorporating sinusoidal PE has no effect on the functional equivalence analysis, and the equivalence classes remain exactly the same as in the absence of positional encoding.

## 3.2 The Setting of Relative Positional Encoding

**Rotary Positional Encoding.** We turn to the *Rotary Positional Encoding* (RoPE) (Su et al., 2024). For each token position $n$, we specify the block-diagonal rotation matrix $R_n \in \mathbb{R}^{d_h \times d_h}$ by

$$R_n = \text{diag}\left(\begin{bmatrix} \cos(n\varphi_1) & -\sin(n\varphi_1) \\ \sin(n\varphi_1) & \cos(n\varphi_1) \end{bmatrix}, \ldots, \begin{bmatrix} \cos(n\varphi_{d_h/2}) & -\sin(n\varphi_{d_h/2}) \\ \sin(n\varphi_{d_h/2}) & \cos(n\varphi_{d_h/2}) \end{bmatrix}\right), \tag{6}$$

where $\varphi_i = 10000^{-2(i-1)/d}$ for $i \in [d_h/2]$. We omit, for clarity, the explicit subscript for the head size $d_h$. Noting that $R_n = (R_1)^n$, the multihead attention with RoPE takes the form

$$\text{MHA}_{\text{RoPE}}(\mathbf{x}: \theta) = \sum_{i=1}^{h} \text{softmax}\left(\left(\mathbf{x}W_i^Q R_m\right)\left(\mathbf{x}W_i^K R_n\right)^\top\right) \cdot \left(\mathbf{x}W_i^V\right)(W_i^O)^\top$$

$$= \sum_{i=1}^{h} \text{softmax}\left[x_m W_i^Q R_{m-n}(W_i^K)^\top x_n^\top\right]_{m,n=1,\ldots,L} \cdot \mathbf{x}W_i^V (W_i^O)^\top. \tag{7}$$

**Analysis of RoPE in Relation to Internal Structure and Symmetry.** The parameterization and parameter domain of MHARoPE match those of the vanilla MHA, but the action of $G$Att on $\Theta$ is no longer symmetric. Specifically, for $\theta \in \Theta$ and $g \in G_{\text{Att}}$, one generally has $\text{MHA}_{\text{RoPE}}(\cdot; \theta) \ne \text{MHA}_{\text{RoPE}}(\cdot; g\theta)$. The underlying cause is that, while $W_i^V$ and $W_i^O$ still interact multiplicatively as in the vanilla case, $W_i^Q$ and $W_i^K$ are now separated by the relative rotary matrix $R_{m-n}$. This insertion blocks the cancellation of $\text{GL}(d_h)$ group actions, and thus the invariance property fails

**Symmetry Group.** To define the symmetry group, first, for $i \in [d_h/2]$, define matrices $P_i, J_i \in \mathbb{R}^{d_h \times d_h}$, each being block-diagonal with $d_h/2$ consecutive $2 \times 2$ diagonal blocks:

$$P_i = \text{diag}\left(0, \ldots, 0, \underbrace{\begin{bmatrix} 1 & 0 \\ 0 & 1 \end{bmatrix}}_{i\text{-th block}}, 0, \ldots, 0\right), \quad J_i = \text{diag}\left((0, \ldots, 0, \underbrace{\begin{bmatrix} 0 & -1 \\ 1 & 0 \end{bmatrix}}_{i\text{-th block}}, 0, \ldots, 0\right). \tag{8}$$

Now define the following group

$$\text{H}(d_h) := \left\{\{U = \sum_{i=1}^{d_h/2}(a_i P_i + b_i J_i) \in \mathbb{R}^{d_h \times d_h} : (a_i, b_i) \in \mathbb{R}^2 \setminus \{(0,0)\}, i \in [d_h/2]\right\}. \tag{9}$$

Verifying directly, $\text{H}(d_h)$ forms an abelian subgroup of $\text{GL}(d_h)$, and moreover it is isomorphic to $(\mathbb{C}^\times)^{d_h/2}$, where $\mathbb{C}^\times$ denotes the multiplicative group of nonzero complex numbers. In particular, for each $n$, the rotary matrix $R_n$ belongs to $\text{H}(d_h)$. We proceed to define

$$G_{\text{RoPE}} := S_h \times (\text{H}(d_h) \times \text{GL}(d_h))^h. \tag{10}$$

Thus, $G_{\text{RoPE}}$ is clearly a subgroup of $G_{\text{Att}}$. Furthermore, the natural action of $G_{\text{Att}}$ on $\Theta$ restricts to $G_{\text{RoPE}}$, yielding a valid group action on $\Theta$. Crucially, this action preserves the behavior of $\text{MHA}_{\text{RoPE}}$, so that $G$RoPE forms a symmetry group of $\text{MHA}_{\text{RoPE}}$.

**Remark 3.1.** The argument proceeds as follows. In comparison with the vanilla MultiHead map, aside from the head permutation $\sigma$ and the product structure of $W_i^V$ and $W_i^O$, the only modification

concerns the interaction of $W_i^Q$ with $W_i^K$. Using the fact that $\mathrm{H}(d_h)$ is abelian and that $R_n \in \mathrm{H}(d_h)$, we obtain

$$
\begin{aligned}
(W_i^Q U^\top) R_n (W_i^K U^{-1})^\top &= W_i^Q U^\top R_n (U^{-1})^\top (W_i^K)^\top \\
&= W_i^Q R_n U^\top (U^{-1})^\top (W_i^K)^\top = W_i^Q R_n (W_i^K)^\top. \quad (11)
\end{aligned}
$$

Thus, the product inside the softmax of the MultiHead$_\text{RoPE}$ map is invariant under $G_\text{RoPE}$.

We now show that $G_\text{RoPE}$ constitutes a maximal symmetry group of MHA$_\text{RoPE}$.

**Theorem 3.2** (Maximality of $G_\text{RoPE}$). *Consider two* MHA$_\text{RoPE}$ *maps with $h$ heads, parameterized by $\theta = (W_i^Q, W_i^K, W_i^V, W_i^O)_{i=1}^h$ and $\bar{\theta} = (\bar{W}_i^Q, \bar{W}_i^Q, \bar{W}_i^V, \bar{W}_i^O)_{i=1}^{\bar{h}}$, respectively. Assume that*

1. *In the initial* MHA$_\text{RoPE}$ *map, the $h$ families listed below contain only nonzero matrices,*

$$
\left\{ W_i^Q (W_i^K)^\top + W_i^K (W_i^Q)^\top; \{W_i^Q R^n (W_i^K)^\top\}_{n \in \mathbb{Z}, n \neq 0} \right\}, \text{ for } i \in [h],
$$

   *and these form $h$ mutually distinct families. An analogous condition applies to the second map.*

2. *All matrices $W_i^Q, W_i^K, W_i^V, W_i^O$ and $\bar{W}_i^Q, \bar{W}_i^K, \bar{W}_i^V, \bar{W}_i^O$, for all feasible $i$, are of rank $d_h$.*

*If the two* MHA$_\text{RoPE}$ *maps are identical, then there exists $g \in G$ such that $\bar{\theta} = g\theta$.*

The proof of Theorem 3.2 is provided in Appendix B. Since the proof is lengthy and relies on several key lemmas, we outline the main steps here. First, MHA$_\text{RoPE}$ is reformulated in the form of an exponential polynomial, and techniques from this area are applied to derive relations among the parameters. Next, a structural property of the rotary matrix, established in Lemma B.12 of Appendix B.6, is used to refine the analysis of these relations. Finally, this refinement enables us to recover the existence of the group elements that connect the two parameter sets.

**Remark 3.3.** As $\mathrm{H}(d_h)$ is significantly smaller than $\mathrm{GL}(d_h)$, the expressive class of MHA$_\text{RoPE}$ strictly exceeds that of MHA or MHA$_\text{APE}$. *This observation gives theoretical support for the widespread use of RoPE in attention models.*

## 4 TELEPORTATION VIA MINIMAL PERTURBATION

In this section, we explore the integration of teleportation techniques into optimization methods.

Given a parameterized function $f(\cdot; \theta)$ with $\theta \in \Theta$, let $G$ be a symmetry group of $f$. Our goal is to minimize the loss function $\mathcal{L}(\theta)$. During optimization, at teleportation steps $K \subseteq \{0, \ldots, T-1\}$, prior work uses expensive Hessian-based methods (Zhao et al., 2022a; Mishkin et al., 2024) to find an optimal $g \in G$. Such methods suffer from high memory costs (Nilsen et al., 2019) and numerical instability (Etmann, 2019). Instead, we propose a simpler, sampling-based alternative.

While weight perturbations that increase the gradient norm can improve performance (Hochreiter & Schmidhuber, 1997; Armenta et al., 2023), the underlying mechanism involves large transformations that alter gradient dynamics. Such drastic changes, even when loss-preserving, risk moving the optimizer into unfavorable regions and impairing convergence and generalization.

In contrast, we argue that small perturbations alongside a standard optimizer (e.g., SGD or Adam) promote faster convergence. Small perturbations keep the optimization trajectory aligned with the optimizer's guidance, avoiding disruptive shifts. This approach balances the exploration from teleportation with the stability required for efficient convergence. Formally, let the symmetry group $G$ be equipped with a metric $d_G$. We define the ball of radius $\alpha > 0$ around the identity $\mathrm{id}_G$ as:

$$
B_G(\alpha) := \{g \in G : d_G(g, \mathrm{id}_G) < \alpha\}. \quad (12)
$$

Each teleportation step is now performed within this ball $B_G(\alpha)$, ensuring that the applied transformation remains within a controlled perturbation range. Therefore, the optimal $g$ is given by:

$$
g \leftarrow \underset{g \in B_G(\alpha)}{\mathrm{argmax}} \|(\nabla \mathcal{L})|_{g\theta}\|_2. \quad (13)
$$

To avoid the prohibitive cost of solving the intractable optimization in Eq. (13), we adopt a sampling-based approach to update $g$. With a fixed budget of $M$ samples, the teleportation update is:

$$g \leftarrow \underset{i=1,\ldots,M}{\operatorname{argmax}}\big\{\|(\nabla\mathcal{L})|_{g_1\theta}\|_2, \ldots, (\nabla\mathcal{L})|_{g_M\theta}\|_2\big\}. \tag{14}$$

For the general linear group $\mathrm{GL}(n)$–a metric space whose metric is induced from the space of $n \times n$ matrices–we sample near the identity by constructing a diagonal matrix as follows $\mathrm{diag}(x_1, \ldots, x_n)$, where each diagonal entry $x_i$ is sampled from $\mathcal{U}([1-\alpha, 1+\alpha])$. This creates controlled perturbations near the identity matrix. Furthermore, if the current parameters $\theta_t$ already have a high gradient norm compared to their symmetric neighbors, they are likely in a favorable optimization region. Further teleportation could then create an excessively large gradient, pushing the optimizer into an unstable region of the loss (Zhao et al., 2022a; Mishkin et al., 2024). To mitigate this risk, we impose a stability condition: teleportation is applied only if a majority of samples increase the gradient norm. Let $\mathcal{S}_t$ be the set of such samples:

$$\mathcal{S}_t = \{g \in \{g_1, \ldots, g_M\} : \|\nabla\mathcal{L}|_{g\theta_t}\|_2 > \|\nabla\mathcal{L}|_{\theta_t}\|_2\}. \tag{15}$$

The update rule for $g$ becomes

$$g = \begin{cases} \underset{g \in \{g_i\}}{\operatorname{argmax}} \|\nabla\mathcal{L}|_{g\theta}\|_2 & \text{if } |\mathcal{S}_t| > M/2, \\ \mathrm{id}_G & \text{otherwise.} \end{cases} \tag{16}$$

The parameters are updated via $\theta \leftarrow g\theta$ before the standard optimizer step. Our full algorithm, Teleportation Training with Sampling Minimal Perturbations, is summarized in Algorithm 1.

**Remark 4.1.** Note that, since the action of the permutation group $S_n$ commutes with the summation operator in $\|(\nabla\mathcal{L})|_{g\theta}\|_2$, its effect does not influence the optimization process. As a result, we can disregard the permutation symmetry, and focus on groups that are equipped with a metric.

## 5 EXPERIMENTS

This section provides an evaluation of our approach on a set of vision and NLP benchmarks. We conduct experiments with multiple architectures and PE schemes, including APE and RoPE, demonstrating the flexibility and general applicability of the proposed framework across diverse settings.

### 5.1 EXPERIMENTAL SETUP

**Optimizer Consideration.** We mainly use SGD, as teleportation yields stronger gains in stability and generalization compared to adaptive methods like Adam, where improvements are marginal. SGD also avoids pathologies of adaptive optimizers, such as overfitting small-scale patterns and slower convergence (Appendix D). For completeness, Adam results are also reported with detailed analyses in Appendix F.

**Datasets and Models.** For vision tasks, we adopt the Vision Transformer (ViT) (Dosovitskiy et al., 2020) on MNIST (LeCun et al., 1998), CIFAR-10 (Krizhevsky et al., 2009), and ImageNet-1K (Deng et al., 2009). For language modeling,, we employ Transformer-XL (Dai et al., 2019) on WikiText-103 (Merity et al., 2016). All models are trained with SGD, momentum, and a cosine scheduler. We also compare teleportation under two widely used forms of APE and RoPE. The complete set of hyperparameters is provided in Appendix E, while Table 1 and Figure 2 present the benchmark results obtained with teleportation.

**Teleport Configuration.** We employ Algorithm 1 to implement teleportation, with key hyperparameters including the number of samples $M$, the radius $\alpha$, and the the set of teleportation steps $K$. In addition, we introduce a parameter consecutive steps which specifies how many teleportation steps are applied consecutively. Guidelines for these parameter selection are provided in Appendix G, where we also present an ablation study to highlight their impact on performance in Section 5.3.

### 5.2 EXPERIMENTAL RESULTS

**Overall, teleportation consistently accelerates convergence on both APE and RoPE.** On small datasets such as MNIST and CIFAR-10, training reaches baseline performance 25–60% faster (8–15

Table 1: Performance of models with and without teleportation on MNIST, CIFAR-10, ImageNet-1K (validation accuracy) and WikiText-103 (test perplexity) under different positional encodings. We also compare with the teleportation method of Zhao et al. (2023) on MNIST and CIFAR-10. *Speedup* denotes the relative training reduction needed for teleportation to match the baseline; *N/A* indicates no measurable improvement.

| Dataset | Teleport | APE | | | RoPE | | |
|---|---|---|---|---|---|---|---|
| | | Accuracy (%) ↑ & PPL ↓ | Speedup (%) ↑ | Time/epoch ↓ | Accuracy (%) ↑ & PPL ↓ | Speedup (%) ↑ | Time/epoch ↓ |
| MNIST | No | $98.03 \pm 0.12$ | - | $8.07 \pm 0.32$s | $97.80 \pm 0.10$ | - | $8.10 \pm 0.19$s |
| | Yes | $\mathbf{98.38 \pm 0.15}$ | $43.41 \pm 9.94$ | $8.24 \pm 0.35$s | $\mathbf{98.41 \pm 0.20}$ | $58.98 \pm 6.35$ | $8.32 \pm 0.25$s |
| | Yes (Zhao) | $97.71 \pm 0.17$ | N/A | $8.36 \pm 0.11$s | $97.82 \pm 0.17$ | $19.37 \pm 9.56$ | $8.30 \pm 0.18$s |
| CIFAR-10 | No | $73.80 \pm 0.44$ | - | $6.97 \pm 0.57$s | $72.58 \pm 0.86$ | - | $6.97 \pm 0.58$s |
| | Yes | $\mathbf{75.44 \pm 0.61}$ | $26.41 \pm 9.48$ | $7.00 \pm 0.28$s | $\mathbf{75.04 \pm 0.88}$ | $29.07 \pm 8.92$ | $7.11 \pm 0.52$s |
| | Yes (Zhao) | $73.69 \pm 0.72$ | N/A | $7.04 \pm 0.09$s | $73.16 \pm 0.18$ | $7.73 \pm 4.63$ | $7.04 \pm 0.06$s |
| ImageNet-1K | No | $67.85 \pm 0.15$ | - | $14.28 \pm 0.03$m | $70.73 \pm 0.07$ | - | $16.47 \pm 0.03$m |
| | Yes | $\mathbf{69.01 \pm 0.23}$ | $17.21 \pm 1.12$ | $14.30 \pm 0.05$m | $\mathbf{71.33 \pm 0.19}$ | $11.65 \pm 1.14$ | $16.50 \pm 0.06$m |
| WikiText-103 | No | $35.76 \pm 0.00$ | - | $15.68 \pm 0.04$m | $36.12 \pm 0.00$ | - | $16.43 \pm 0.05$m |
| | Yes | $\mathbf{35.15 \pm 0.17}$ | $18.73 \pm 1.57$ | $15.72 \pm 0.04$m | $\mathbf{35.70 \pm 0.26}$ | $21.12 \pm 1.85$ | $16.46 \pm 0.04$m |

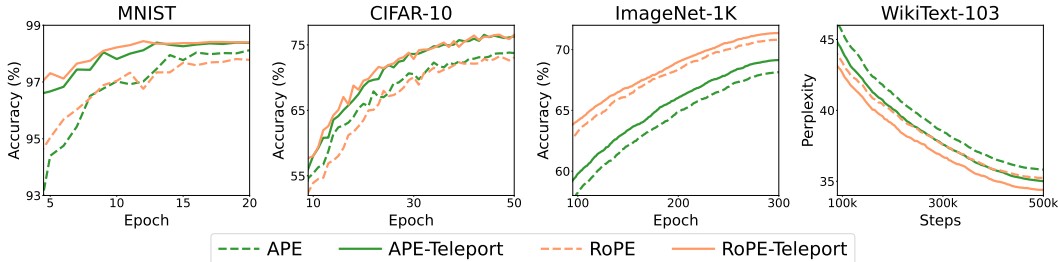

Figure 2: Validation performance on MNIST, CIFAR-10, ImageNet-1K (accuracy) and WikiText-103 (perplexity), comparing models trained with and without teleportation under different PE.

epochs), while larger-scale tasks show more moderate yet substantial gains, 10–18% on ImageNet-1K and about 20% on WikiText-103. RoPE exhibits the most consistent advantage, with acceleration on MNIST reaching 59% versus 43% under APE. Beyond speed, improvements in final accuracy and perplexity are modest (e.g., 2.46% on CIFAR-10), and the runtime overhead per epoch remains negligible. These results highlight teleportation as a practical approach to reducing convergence time in both vision and language models without sacrificing generalization.

**Comparing algorithms.** We compare our method with Zhao et al. (2023) on MNIST and CIFAR-10. Our approach incurs a 2% computational overhead, whereas Zhao's requires about double the GPU memory (Table 3) without significant gains in accuracy or efficiency (Table 1, Appendix E).

### 5.3 ABLATION STUDY

This section investigates the sensitivity of teleportation to different configuration choices through an ablation study. The complete ablation study results are reported in Table 2.

**Datasets and Models.** We conduct ablations on CIFAR-10 and WikiText-103 using RoPE (details of architectures and hyperparameters in Appendix E). On CIFAR-10, we vary teleportation settings across attention layers, radius, number of teleportation steps, teleportation epochs, and FFN contribution. On WikiText-103, we analyze how teleportation step positions affect convergence speed.

**Attention layers and FFN.** Teleporting only the first attention layer hurts performance, while applying it to the last layer improves it; teleporting all layers achieves the best results. In contrast, combining Attention and FFN often underperforms the baseline.

**Radius and Number of Steps.** Smaller radius or step counts yield weaker results, but overly large values destabilize training. A balanced trade-off is required, where a smaller radius can be offset by more steps and vice versa.

Table 2: Ablation results on CIFAR-10 with RoPE, varying the teleported attention layers, teleportation radius $\alpha$, number of teleportation steps $|K|$, teleportation epochs, and FFN contribution. Results are reported as the mean and standard deviation over five runs. *N/A* indicates that the improvement in training time cannot be measured because the validation accuracy does not surpass the non-teleportation baseline.

| Change | Layers | $\alpha$ | $|K|$ | Epochs | FFN | Val Acc (%) ↑ | Speedup (%) ↑ | Change | Layers | $\alpha$ | $|K|$ | Epochs | FFN | Val Acc (%) ↑ | Speedup (%) ↑ |
|---|---|---|---|---|---|---|---|---|---|---|---|---|---|---|---|
| Layers | first | 0.65 | 4 | 1 | 0 | $69.65 \pm 1.24$ | N/A | $|K|$ | all | 0.65 | 2 | 1 | 0 | $73.24 \pm 1.30$ | $13.47 \pm 11.42$ |
| | last | 0.65 | 4 | 1 | 0 | $74.04 \pm 0.16$ | $19.31 \pm 0.25$ | | all | 0.65 | 4 | 1 | 0 | $75.04 \pm 0.88$ | $29.07 \pm 8.92$ |
| | all | 0.65 | 4 | 1 | 0 | $\mathbf{75.04 \pm 0.88}$ | $29.07 \pm 8.92$ | | all | 0.65 | 6 | 1 | 0 | $\mathbf{75.70 \pm 0.34}$ | $31.46 \pm 0.37$ |
| $\alpha$ | all | 0.9 | 4 | 1 | 0 | $66.17 \pm 7.92$ | N/A | | all | 0.65 | 8 | 1 | 0 | $73.26 \pm 1.71$ | $22.90 \pm 3.04$ |
| | all | 0.65 | 4 | 1 | 0 | $\mathbf{75.04 \pm 0.88}$ | $29.07 \pm 8.92$ | | all | 0.65 | 10 | 1 | 0 | $67.47 \pm 6.35$ | N/A |
| | all | 0.5 | 4 | 1 | 0 | $73.74 \pm 0.18$ | $17.94 \pm 1.59$ | Epochs | all | 0.65 | 4 | 1 | 0 | $75.04 \pm 0.88$ | $29.07 \pm 8.92$ |
| | all | 0.5 | 8 | 1 | 0 | $74.70 \pm 1.44$ | $30.85 \pm 11.66$ | | all | 0.65 | 4 | 3 | 0 | $\mathbf{75.17 \pm 1.11}$ | $33.73 \pm 4.15$ |
| | all | 0.3 | 8 | 1 | 0 | $71.77 \pm 1.34$ | N/A | | all | 0.4 | 8 | 1, 2 | 0 | $75.08 \pm 0.81$ | $24.08 \pm 8.72$ |
| | all | 0.3 | 16 | 1 | 0 | $75.00 \pm 1.40$ | $36.62 \pm 3.01$ | | all | 0.3 | 8 | 1, 2, 3 | 0 | $73.94 \pm 0.81$ | $24.50 \pm 4.80$ |
| FFN | all | 0.65 | 4 | 1 | 1 | $70.89 \pm 1.62$ | N/A | | all | 0.3 | 8 | 1, 3, 5 | 0 | $74.49 \pm 0.38$ | $24.13 \pm 3.81$ |

**Teleportation epochs (steps).** The effectiveness of teleportation depends strongly on when it is applied. On CIFAR-10, spreading teleportation across multiple epochs forces reductions in step count or radius to prevent gradient explosion, yielding weaker results than concentrating it at a single well-chosen epoch with a larger radius. Similar sensitivity is observed on WikiText-103 (Table 6), optimal performance arises when teleportation occurs during an intermediate warmup stage (25–50%), where gradients are sufficiently scaled, before stabilized convergence is reached.

**Training time.** Increasing the sample size $M$ improves stability but adds runtime, with theoretical overhead $\sim 100 \cdot \frac{M \cdot |K|}{\text{total steps}}\%$. As $M$ and $|K|$ are typically small(Appendix G), the cost remains below 3%, while practical system-level variability rarely causes significant slowdowns.

## 5.4 GENERALIZATION

Beyond its impact on convergence speed, teleportation also enhances the generalization.

**Teleportation converges to flatter minima.** While our primary goal is to amplify gradient magnitudes, we also observe improved validation accuracy, suggesting enhanced generalization. Sharpness analysis following Foret et al. (2020) confirms that teleportation leads to flatter minima (Table 4), consistent with prior findings (Zhao et al., 2023).

**Large noise of gradient.** Complementary evidence arises from gradient noise estimation using the methodology of Wu et al. (2020), which reveals elevated noise levels after teleportation (Figure 3a). This observation agrees with prior findings Smith & Le (2017); Feng & Tu (2021), which argue that increased stochastic gradient noise can promote better generalization.

**Smaller $\ell_2$ gradient norms.** We additionally analyze the dynamics of $\ell_2$ gradient norms throughout training. Teleportation produces larger norms in the early stages but smaller ones toward the end (Figure 3b). This pattern resonates with the insights of Zhao et al. (2022b), which demonstrate that reduced gradient magnitudes in later phases are conducive to stronger generalization.

Taken together, these results suggest that teleportation not only accelerates optimization but also implicitly enhances generalization by promoting flatter minima, injecting beneficial gradient noise, and shaping gradient dynamics in a favorable manner.

## 6 CONCLUSION

In this paper, we introduce a framework for functional equivalence, symmetry groups, and maximal symmetry groups. We analyze Multihead Attention with a focus on how positional encodings reshape the symmetry structure of vanilla attention—a perspective not formally addressed before. Building on this, we propose a teleportation-based method to accelerate Transformer optimization. Experiments demonstrate that teleportation improves both convergence speed and model performance, and we further identify suitable configurations across datasets of different scales. However, threshold selection remains limited, and the behavior of teleportation on very large models such as LLMs has yet to be explored, which we highlight as an important direction for future work.

**Ethics Statement.** Due to its emphasis on technical and methodological elements, this research does not present any anticipated risks of harmful societal or ethical effects.

**Reproducibility Statement.** The full source code for all experiments is supplied in the supplementary materials. Information on hyperparameters, training procedures, and computing resources is outlined in Appendix E. All datasets utilized in this study are openly accessible and readily available online.

**LLM Usage Declaration.** Large language models (LLMs) were used exclusively for proofreading grammar and making slight linguistic adjustments.

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

TABLE OF NOTATION

---

*General Mathematical Notation*

| | |
|---|---|
| $\mathbb{R}^n$ | $n$-dimensional Euclidean space |
| $\mathbb{R}^{m \times n}$ | Space of $m \times n$ real matrices |
| $\text{softmax}(\cdot)$ | Row-wise softmax operator |
| $\|\cdot\|_2$ | Euclidean norm (for vectors or gradients) |

*Dimensions and Indices*

| | |
|---|---|
| $d$ | Dimension of token embeddings |
| $d_h$ | Dimension of each attention head |
| $h$ | Number of attention heads in a model |
| $L$ | Length of the input token sequence |
| $m, n, k$ | Indices representing positions in a sequence or shifts |
| $i$ | Index representing attention heads |

*Spaces and Parameters*

| | |
|---|---|
| $\mathcal{S}$ | The space of all token sequences, $\bigsqcup_{L=1}^{\infty} \mathbb{R}^{L \times d}$ |
| $W_i^Q, W_i^K, W_i^V, W_i^O$ | Query, key, value, and output matrices of head $i$, each in $\mathbb{R}^{d \times d_h}$ |
| $\theta$ | The complete set of parameters for a multi-head attention layer |
| $\Theta$ | The parameter space for a multi-head attention layer, $(\mathbb{R}^{d \times d_h})^{4h}$ |
| $A_i^{m,n}, B_i$ | Parameter matrices for the general multi-head attention formulation |

*Symmetry Groups*

| | |
|---|---|
| $S_h$ | The permutation group on a set of $h$ elements |
| $\text{GL}(d_h)$ | The general linear group of invertible $d_h \times d_h$ matrices |
| $G_{\text{Att}}$ | The symmetry group for standard multi-head attention |
| $g$ | Element of $G_{\text{Att}}$, $g = (\sigma, (U_i, V_i)_{i=1}^h)$ with $\sigma \in S_h, U_i, V_i \in \text{GL}(d_h)$ |

*Positional Encodings (RoPE)*

| | |
|---|---|
| $R^n$ | The block-diagonal rotation matrix for relative position $n$ in RoPE |
| $\theta_i$ | The rotation angle (frequency) for the $i$-th 2D block in RoPE matrices |
| $P_i, J_i$ | Projection and skew-symmetric matrices for the $i$-th 2D block |

*Optimization and Teleportation*

| | |
|---|---|
| $\mathcal{L}(\theta)$ | Loss function to minimize |
| $\nabla \mathcal{L}\|_\theta$ | Gradient of the loss at parameters $\theta$ |
| $\varphi$ | Optimizer update function |
| $T$ | Total number of optimization steps |
| $K$ | Set of teleportation steps |
| $\alpha > 0$ | Perturbation range for sampling |
| $M$ | Number of samples for teleportation |
| $B_G(\alpha)$ | Ball of radius $\alpha$ in $G$ centered at the identity, w.r.t. metric $d_G$ |

---

# Supplement to "Accelerating Transformer Training: Architectural Symmetry, Positional Encoding, and Teleportation"

## Table of Contents

## A  FUNCTIONAL EQUIVALENCE OF VANILLA MULTIHEAD ATTENTION

Let $d, d_h$ be positive integers with $d \geq d_h$. A multihead attention operator with $h$ heads is defined by

$$\mathrm{MHA}\Big(\mathbf{x}; \{W_i^Q, W_i^K, W_i^V, W_i^O\}_{i=1}^h\Big)$$

$$= \sum_{i=1}^h \mathrm{softmax}\Big((\mathbf{x}W_i^Q)(\mathbf{x}W_i^K)^\top\Big)\,(\mathbf{x}W_i^V)(W_i^O)^\top, \tag{17}$$

where $W_i^Q, W_i^K, W_i^V, W_i^O \in \mathbb{R}^{d \times d_h}$. The operator is parameterized by

$$\theta := \big(W_i^Q, W_i^K, W_i^V, W_i^O\big)_{i=1}^h, \tag{18}$$

and its parameter space is

$$\Theta := \big(\mathbb{R}^{d \times d_h}\big)^{4h}. \tag{19}$$

For brevity, the number of heads $h$ is omitted from the notation $\Theta$. When it is necessary to emphasize $h$, we write $\Theta_h$.

**Group Action on the Parameter Space.** Define the following group

$$G_{\text{Att}} \coloneqq S_h \times (\text{GL}(d_h) \times \text{GL}(d_h))^h. \tag{20}$$

This is precisely the direct product between the permutation group $S_h$ and $h$ copies of $\text{GL}(d_h) \times \text{GL}(d_h)$. Each group element $g \in G_{\text{Att}}$ has the form

$$g \coloneqq (\sigma, (U_i, V_i)_{i=1}^h), \tag{21}$$

where $\sigma \in S_h$ and $U_i, V_i \in \text{GL}(d_h)$. The natural action of $G_{\text{Att}}$ on the parameter space $\Theta$ is defined by

$$g\theta \coloneqq \left( W_{\sigma(i)}^Q \cdot U_i^\top, W_{\sigma(i)}^K \cdot U_i^{-1}, W_{\sigma(i)}^V \cdot V_i^\top, W_{\sigma(i)}^O \cdot V_i^{-1} \right)_{i=1}^h \tag{22}$$

This action preserves the functionality of the MHA map: for all $\theta \in \Theta$ and all $g \in G_{\text{Att}}$,

$$\text{MHA}(\cdot; \theta) = \text{MHA}(\cdot; g\theta). \tag{23}$$

The contribution of the general linear group action vanishes through cancellation in the matrix multiplications, while the action induced by the permutation $\sigma$ commutes with the addition operator. Taken together, these actions characterize the full symmetry of the multihead attention mechanism, as established in the following result from Tran et al. (2025).

**Theorem A.1** (See Tran et al. (2025)). *Let*

$$\theta = \left( W_i^Q, W_i^K, W_i^V, W_i^O \right)_{i=1}^h \in \Omega_h, \ \ and \ \ \bar{\theta} = \left( \bar{W}_i^Q, \bar{W}_i^K, \bar{W}_i^V, \bar{W}_i^O \right)_{i=1}^{\bar{h}} \in \Omega_{\bar{h}}, \tag{24}$$

*be two parameterizations of* MHA *maps. Assume that:*

1. *Every $d \times d_h$ matrix appearing in $\theta$ and $\bar{\theta}$ has full column rank $d_h$;*

2. *The matrices $\{W_i^Q (W_i^K)^\top\}_{i=1}^h$ are pairwise distinct;*

3. *The matrices $\{\bar{W}_i^Q (\bar{W}_i^K)^\top\}_{i=1}^{\bar{h}}$ are pairwise distinct.*

*If the two MHA maps are identical, i.e.,*

$$\text{MHA}(\cdot; \theta) = \text{MHA}(\cdot; \bar{\theta}), \tag{25}$$

*then, $h = \bar{h}$, and there exists $g \in G_{\text{Att}}$ such that $\bar{\theta} = g\theta$.*

# B   A PROOF FOR THEOREM 3.2 AND A GENERALIZED VERSION

## B.1   A GENERAL FORMULATION FOR MULTIHEAD ATTENTION

Consider an $h$-head MHA, specified by the following parameters:

$$\theta = \{\{A_i^{m,n}\}_{m,n}, B_i\}_{i=1}^h, \tag{26}$$

where every $A_i^{m,n}$ and $B_i$ are elements of $\mathbb{R}^{d \times d}$, specified as:

$$\text{MHA}(\mathbf{x}; \theta) = \sum_{i=1}^h \text{softmax} \left[ x_m A_i^{m,n} x_n^\top \right]_{m,n=1,\dots,L} \cdot \mathbf{x} B_i. \tag{27}$$

The subsequent analysis of the general MHA is preceded by two preliminary observations.

1. For all integers $m, n \geq 1$ and shifts $k \geq 0$, we take

$$A^{m,n} = A^{m+k,n+k}. \tag{28}$$

This aligns with the natural stationarity constraint enforced by relative positional encodings.

2. For each $m \geq 1$, the similarity of the $m$-th token with itself at head $i$ is computed by a function $f$ parameterized by $A_i^{m,m}$, namely

$$x_m A_i^{m,m} x_m^\top. \tag{29}$$

Given that any quadratic form uniquely corresponds to a symmetric matrix, we may assume $A_i^{m,m}$ is symmetrized:

$$A_i^{m,m} \longmapsto \frac{A_i^{m,m} + (A_i^{m,m})^\top}{2}. \tag{30}$$

This transformation keeps the function unchanged:

$$x_m A_i^{m,m} x_m^\top = x_m \left( \frac{A_i^{m,m} + (A_i^{m,m})^\top}{2} \right) x_m^\top. \tag{31}$$

Thus, going forward, we suppose that $A_i^{m,m}$ is symmetric for all $i, m$.

We now turn to the case, under this framework, where two MHA maps with $h$ and $\bar{h}$ heads produce equivalent outputs:

$$\text{MHA}(\mathbf{x}; \theta) = \text{MHA}(\mathbf{x}; \bar{\theta}). \tag{32}$$

From $g(\cdot : B) = -g(\cdot : -B)$, it follows that Equation (32) amounts to asserting that a MultiHead map with $h + \bar{h}$ heads vanishes everywhere:

$$0 = \text{MHA}(\mathbf{x}; \theta \sqcup \bar{\theta}). \tag{33}$$

The analysis of functional equivalence begins with identifying when a MultiHead map is identically zero. Prior to presenting the proof, we put forth the following definition. Two parameter families $\{A^{m,n}\}_{m,n \geq 1}$ and $\{\bar{A}^{m,n}\}_{m,n \geq 1}$ are said to be *distinct* provided there exist indices $m, n \geq 1$ for which

$$A^{m,n} \neq \bar{A}^{m,n}. \tag{34}$$

The stage is now set to introduce the main theorem of this section.

## B.2 FUNCTIONAL EQUIVALENCE OF GENERAL MULTIHEAD ATTENTION

**Theorem B.1** (Linear independence in general MHA). *We focus on the* MultiHead *operator with $h$ heads, parameterized by $\theta$, under the assumption that the parameter families*

$$\{A_1^{m,n}\}_{m,n \geq 1}, \{A_2^{m,n}\}_{m,n \geq 1}, \ldots, \{A_h^{m,n}\}_{m,n \geq 1}, \tag{35}$$

*are mutually distinct, with the condition that $A_i^{m,n}$ is nonzero for each $i \in [h]$ and every $m, n \geq 1$. If, for all $\mathbf{x} \in \mathcal{S} = \sqcup_{L=1}^{\infty} \mathbb{R}^{L \times d}$, , the following holds:*

$$\text{MHA}(\mathbf{x}; \theta) = 0. \tag{36}$$

*then, $B_1, \ldots, B_h$ are equal to $0$.*

*Proof.* To aid understanding, we outline the principal steps of the proof at a high level:

1. **Preliminary setup.** To set the stage for the proof, we begin with a few preliminary remarks and notational conventions. The argument reduces to showing that at least one coefficient $B_i$ vanishes. Symmetry in the setup then guarantees that every $B_i$ must be zero, proving the theorem.

2. **Reformulation as an exponential polynomial.** From Equation (36), we obtain

$$0 = \sum_{(t_1, \ldots, t_h) \in [L]^h} \exp\left( \sum_{i=1}^{h} x_k A_i^{k,t_i} x_{t_i}^\top \right) \left( \sum_{i=1}^{h} x_{t_i} B_i \right). \tag{37}$$

By a double-counting argument, this identity holds. The corresponding expression forms an exponential polynomial that is everywhere zero. To proceed, we invoke linear independence results for exponential functions over rational fields, which force specific relations among the coefficients.

3. **Structural constraints on the $B_i$.** Using the linear independence principle, we deduce a key structural restriction on the coefficients $B_i$. In particular, the symmetry conditions imposed by the $A_i^{k,t}$ on permissible permutations enforce a collection of linear relations among the $B_i$, indexed by $i \in [h]$. These relations lie at the heart of the proof: they reduce the analysis of a complex exponential sum to checking the consistency of a system of linear equations in the $B_i$.

4. **Partition-based refinement.** Next, we investigate the equalities arising among the families $\{A_i^{k,t}\}_{i=1}^h$. This step clarifies that the structural relations from the previous stage are both necessary and sufficient to ensure that at least one $B_i$ must vanish. The refinement makes use of the partitioning $\{U_p\}$ together with carefully chosen subsets $V^{t_j}$, allowing us to sharpen the constraints and identify the relevant indices.

5. **Conclusion.** In the final step, we integrate the arguments developed above. The structural relations identified in **Step 3**, once refined through the partition analysis of **Step 4**, ensure that at least one $B_i$ must vanish. By the reduction carried out in Step 1, it follows that every $B_i$ is zero, which completes the theorem.

The complete argument is presented as follows.

**Step 1.**

We express the formulation of

$$\mathrm{MHA}(\mathbf{x}; \theta) \tag{38}$$

in a token-wise manner. From Equation (36), for every $1 \leq k \leq L$, one has

$$\sum_{i=1}^h \left( \sum_{j=1}^L \frac{\exp(x_k A_i^{k,j} x_j^\top)}{\sum_{q=1}^L \exp(x_k A_i^{k,q} x_q^\top)} \cdot x_j B_i \right) = 0. \tag{39}$$

Since the families $\{A_1^{m,n}\}_{m,n\geq 1}, \{A_2^{m,n}\}_{m,n\geq 1}, \ldots, \{A_h^{m,n}\}_{m,n\geq 1}$ are pairwise distinct, and for each $i$, $A_i^{m,n}$ depends only on the difference $(m-n)$, one can choose a sufficiently large $L$ and an index $k$ such that the $h$ sets

$$\{A_1^{k,n}\}_{n\geq 1}, \ \{A_2^{k,n}\}_{n\geq 1}, \ \ldots, \ \{A_h^{k,n}\}_{n\geq 1}$$

are pairwise distinct. For the remainder of the proof, we fix such a $k$ and consider all $L \geq k$.

By induction, it suffices to establish that at least one of $B_1, \ldots, B_h$ vanishes. Indeed, if this holds, then the problem reduces to a MultiHead Attention mechanism with fewer heads, and repeating the argument shows that all $B_1, \ldots, B_h$ must be zero. Consequently, our goal is to prove that there exists at least one index $1 \leq i \leq h$ such that $B_i = 0$.

**Step 2.**

First, we rewrite Equation (39) in a more convenient form. By multiplying out all denominators in Equation (39), we obtain

$$\sum_{i=1}^h \left( \sum_{j=1}^L \exp\left( x_k A_i^{k,j} x_j^\top \right) \cdot \prod_{p \in [h] \setminus \{i\}} \left( \sum_{q=1}^L \exp\left( x_k A_p^{k,q} x_q^\top \right) \right) \cdot x_j B_i \right) = 0. \tag{40}$$

We now observe that the left-hand side of Equation (40) can be re-expressed as

$$\sum_{i=1}^h \left( \sum_{j=1}^L \exp\left( x_k A_i^{k,j} x_j^\top \right) \cdot \prod_{p \in [h] \setminus \{i\}} \left( \sum_{q=1}^L \exp\left( x_k A_p^{k,q} x_q^\top \right) \right) \cdot x_j B_i \right)$$

$$= \sum_{(t_1, \ldots, t_h) \in [L]^h} \exp\left( \sum_{i=1}^h x_k A_i^{k,t_i} x_{t_i}^\top \right) \left( \sum_{i=1}^h x_{t_i} B_i \right). \tag{41}$$

To verify Equation (41), define for $i \in [h]$ and $j \in [L]$,

$$a_{i,j} := \exp\left( x_k A_i^{k,j} x_j^\top \right), \qquad b_{i,j} := x_j B_i. \tag{42}$$

In this notation, the claimed identity becomes

$$\sum_{i=1}^h \left( \sum_{j=1}^L a_{i,j} \prod_{p \in [h] \setminus \{i\}} \sum_{q=1}^L a_{p,q} \cdot b_{i,j} \right) = \sum_{(t_1,\dots,t_h) \in [L]^h} \left( \prod_{i=1}^h a_{i,t_i} \right) \left( \sum_{i=1}^h b_{i,t_i} \right). \tag{43}$$

For $(i, \mathbf{t}) \in [h] \times [L]^h$, define the weight

$$w(i, \mathbf{t}) := \Big( \prod_{p=1}^h a_{p,t_p} \Big) b_{i,t_i}. \tag{44}$$

We will compute the following quantity in two ways,

$$\sum_{(i,\mathbf{t}) \in [h] \times [L]^h} w(i, \mathbf{t}). \tag{45}$$

*Group by the distinguished index $i$.*

Fix $i \in [h]$. Then

$$\sum_{\mathbf{t} \in [L]^h} w(i, \mathbf{t}) = \sum_{t_i=1}^L \sum_{(t_p)_{p \neq i} \in [L]^{h-1}} \Big( \prod_{p=1}^h a_{p,t_p} \Big) b_{i,t_i}$$

$$= \sum_{t_i=1}^L a_{i,t_i} \, b_{i,t_i} \underbrace{\sum_{(t_p)_{p \neq i} \in [L]^{h-1}} \prod_{p \neq i} a_{p,t_p}}_{(\star)}. \tag{46}$$

The inner sum $(\star)$ equals

$$\prod_{p \neq i} \sum_{q=1}^L a_{p,q}, \tag{47}$$

since expanding the product enumerates every choice of $(t_p)_{p \neq i}$ exactly once. Hence

$$\sum_{\mathbf{t} \in [L]^h} w(i, \mathbf{t}) = \sum_{j=1}^L a_{i,j} \Big( \prod_{p \neq i} \sum_{q=1}^L a_{p,q} \Big) b_{i,j}. \tag{48}$$

Summing over $i = 1, \dots, h$ yields the left-hand side of Equation (43).

*Group by the tuple $\mathbf{t}$.*

Fix $\mathbf{t} = (t_1, \dots, t_h) \in [L]^h$. Then

$$\sum_{i=1}^h w(i, \mathbf{t}) = \sum_{i=1}^h \Big( \prod_{p=1}^h a_{p,t_p} \Big) b_{i,t_i} = \Big( \prod_{p=1}^h a_{p,t_p} \Big) \Big( \sum_{i=1}^h b_{i,t_i} \Big). \tag{49}$$

Summing over all $\mathbf{t}$ yields the right-hand side of Equation (43).

In conclusion, both groupings compute the same total $\sum_{(i,\mathbf{t}) \in \Omega} w(i, \mathbf{t})$, so Equation (43) holds. Substituting back $a_{i,j} = \exp(x_k A_i^{k,j} x_j^\top)$, $b_{i,j} = x_j B_i$ recovers the original identity. From Equation (40) and Equation (41), we conclude that

$$0 = \sum_{(t_1,\dots,t_h) \in [L]^h} \left[ \exp\left( \sum_{i=1}^h x_k A_i^{k,t_i} x_{t_i}^\top \right) \left( \sum_{i=1}^h x_{t_i} B_i \right) \right]. \tag{50}$$

Note that in Equation (50), both sides represent vectors in $\mathbb{R}^d$. If we examine a single coordinate of this vector, the identity remains valid by restricting each $B_i$ to the corresponding column indexed by that coordinate. Hence, without loss of generality, we may interpret Equation (50) under the convention that each $B_i$ is regarded as a column vector in $\mathbb{R}^d$ corresponding to the chosen coordinate.

**Step 3.**

For $(t_1, \ldots, t_h) \in \mathbb{N}^h$, define

$$g_{(t_1,\ldots,t_h)}(\mathbf{x}) := \sum_{i=1}^{h} x_k A_i^{k,t_i} x_{t_i}^{\top} \qquad\qquad \in \mathbb{R}[\mathbf{x}], \qquad (51)$$

$$h_{(t_1,\ldots,t_h)}(\mathbf{x}) := \sum_{i=1}^{h} x_{t_i} B_i \qquad\qquad \in \mathbb{R}[\mathbf{x}], \qquad (52)$$

$$f_{(t_1,\ldots,t_h)}(\mathbf{x}) := \exp\big(g_{k,(t_1,\ldots,t_h)}(\mathbf{x})\big) \, h_{(t_1,\ldots,t_h)}(\mathbf{x}). \qquad (53)$$

Then Equation (50) can be rewritten as

$$0 = \sum_{(t_1,\ldots,t_h)\in[L]^h} f_{(t_1,\ldots,t_h)}(\mathbf{x})$$

$$= \sum_{(t_1,\ldots,t_h)\in[L]^h} \exp\big(g_{(t_1,\ldots,t_h)}(\mathbf{x})\big) \, h_{(t_1,\ldots,t_h)}(\mathbf{x}). \qquad (54)$$

Observe that each polynomial $g_{(t_1,\ldots,t_h)} \in \mathbb{R}[\mathbf{x}]$ has constant term equal to zero. By Lemma B.3, Equation (54) implies that, for each $g \in \mathbb{R}[\mathbf{x}]$, grouping together all indices $(t_1, \ldots, t_h)$ such that $g_{(t_1,\ldots,t_h)} = g$ yields

$$0 = \sum_{(t_1,\ldots,t_h)\in[L]^h \; : \; g_{(t_1,\ldots,t_h)}=g} \exp\big(g_{(t_1,\ldots,t_h)}(\mathbf{x})\big) \, h_{(t_1,\ldots,t_h)}(\mathbf{x}), \qquad (55)$$

and since $\exp(g(\mathbf{x}))$ is common to all such terms, we conclude

$$0 = \sum_{(t_1,\ldots,t_h)\in[L]^h \; : \; g_{(t_1,\ldots,t_h)}=g} h_{(t_1,\ldots,t_h)}(\mathbf{x}). \qquad (56)$$

One has the following observation. Consider an arbitrary tuple $(t_1, \ldots, t_h) \in [L]^h$ such that $t_1, \ldots, t_h$ are pairwise distinct. Assume that there exists another tuple $(t_1', \ldots, t_h') \in [L]^h$ satisfying

$$g_{(t_1,\ldots,t_h)} = g_{(t_1',\ldots,t_h')}. \qquad (57)$$

Since all $A_i^{m,n}$ are nonzero and $A_i^{m,m}$ is symmetric, it follows that every polynomial of the form $x_m A_i^{m,n} x_n$ is nonvanishing. Consequently, in $g_{k,(t_1,\ldots,t_h)}$, for each $i \in [h]$, there must exist polynomial terms that involve at least one entry of $x_{t_i}$. (This requirement that the $t_i$'s be pairwise distinct is crucial, as it prevents possible cancellation of terms.) Hence, for each $i \in [h]$, there exists $j \in [h]$ such that $t_i = t_j'$. Moreover, since the $t_i$'s are pairwise distinct, it follows that $(t_1', \ldots, t_h')$ must be a *permutation* of $(t_1, \ldots, t_h)$. From Equation (54) and Lemma B.3, one therefore obtains

$$0 = \sum_{\sigma \in S_h} h_{(t_{\sigma(1)},\ldots,t_{\sigma(h)})}(\mathbf{x}). \qquad (58)$$

It should be emphasized, however, that the condition $(t_1', \ldots, t_h')$ being a permutation of $(t_1, \ldots, t_h)$ is not sufficient, in itself, to guarantee that $g_{(t_1,\ldots,t_h)} = g_{(t_1',\ldots,t_h')}$. To examine this more closely, let $(t_1', \ldots, t_h') = (t_{\sigma(1)}, \ldots, t_{\sigma(h)})$ for some $\sigma \in S_h$. From the assumption $g_{(t_1,\ldots,t_h)} = g_{(t_1',\ldots,t_h')}$, we have

$$\sum_{i=1}^{h} x_k A_i^{k,t_i} x_{t_i}^\top = \sum_{i=1}^{h} x_k A_i^{k,t_{\sigma(i)}} x_{t_{\sigma(i)}}^\top. \tag{59}$$

By reindexing the summation, this is equivalent to

$$\sum_{i=1}^{h} x_k A_i^{k,t_i} x_{t_i}^\top = \sum_{i=1}^{h} x_k A_{\sigma^{-1}(i)}^{k,t_i} x_{t_i}^\top, \tag{60}$$

which in turn is equivalent to requiring that $A_i^{k,t_i} = A_{\sigma^{-1}(i)}^{k,t_i}$ for all $i \in [h]$. This shows explicitly the additional algebraic condition that must hold in order for two permutations to yield the same polynomial $g$. Note that this constitutes a sufficient condition on $\sigma \in S_h$ to ensure that $g_{(t_1,\ldots,t_h)} = g_{(t'_1,\ldots,t'_h)}$ whenever $(t'_1,\ldots,t'_h) = (t_{\sigma(1)},\ldots,t_{\sigma(h)})$.

Accordingly, one deduces

$$0 = \sum_{\sigma \in S_h \ : \ A_j^{k,t_j} = A_{\sigma^{-1}(j)}^{k,t_j} \ \forall j \in [h]} h_{(t_{\sigma(1)},\ldots,t_{\sigma(h)})}(\mathbf{x})$$

$$= \sum_{\sigma \in S_h \ : \ A_j^{k,t_j} = A_{\sigma^{-1}(j)}^{k,t_j} \ \forall j \in [h]} \left( \sum_{i=1}^{h} x_{t_{\sigma(i)}} B_i \right)$$

$$= \sum_{\sigma \in S_h \ : \ A_j^{k,t_j} = A_{\sigma^{-1}(j)}^{k,t_j} \ \forall j \in [h]} \left( \sum_{i=1}^{h} x_{t_i} B_{\sigma^{-1}(i)} \right)$$

$$= \sum_{\sigma \in S_h \ : \ A_j^{k,t_j} = A_{\sigma(j)}^{k,t_j} \ \forall j \in [h]} \left( \sum_{i=1}^{h} x_{t_i} B_{\sigma(i)} \right)$$

$$= \sum_{i=1}^{h} \left( x_{t_i} \cdot \sum_{\sigma \in S_h \ : \ A_j^{k,t_j} = A_{\sigma(j)}^{k,t_j} \ \forall j \in [h]} B_{\sigma(i)} \right). \tag{61}$$

Thus, since the entries $t_1,\ldots,t_h$ are pairwise distinct, the monomials $x_{t_i}$ are linearly independent. It therefore follows that, for each $i \in [h]$, one must have

$$0 = \sum_{\sigma \in S_h \ : \ A_j^{k,t_j} = A_{\sigma(j)}^{k,t_j} \ \forall j \in [h]} B_{\sigma(i)}. \tag{62}$$

Equation (62) encapsulates the key structural constraint on the coefficients $B_i$. It shows that, once the $A_i^{k,t}$'s impose symmetry conditions on admissible permutations, the $B_i$'s must satisfy a family of linear relations indexed by $i \in [h]$. This relation will serve as the main tool in subsequent steps, where we will exploit the partition structure of the $U_p$'s to force specific $B_i$'s to vanish.

**Step 4.**

For each $t \in \mathbb{N}$, define $\{U_p^t\}_{p=1}^{\alpha_t}$ to be the unique partition of $[h]$ such that, for $i, j \in [h]$, one has $A_i^{k,t} = A_j^{k,t}$ if and only if $i$ and $j$ belong to the same set $U_p^t$. Since the number of possible partitions of $\{1,\ldots,h\}$ is finite, there exists a partition $\{U_p\}_{p=1}^{\alpha}$ such that the equality

$$\{U_p^t\}_{p=1}^{\alpha_t} = \{U_p\}_{p=1}^{\alpha} \tag{63}$$

holds for infinitely many values of $t \in \mathbb{N}$. Let $S$ denote the set of all such positive integers $t$.

By reindexing the head indices if necessary, we may assume that

$$U_1 = \{1,\ldots,m\}. \tag{64}$$

Next, observe that since the $h$ sequences

$$\{A_1^{k,n}\}_{n\geq 1}, \quad \{A_2^{k,n}\}_{n\geq 1}, \quad \ldots, \quad \{A_h^{k,n}\}_{n\geq 1} \tag{65}$$

are pairwise distinct, there exists a positive integer $K$ such that the truncated sequences

$$\{A_1^{k,n}\}_{n=1}^K, \quad \{A_2^{k,n}\}_{n=1}^K, \quad \ldots, \quad \{A_h^{k,n}\}_{n=1}^K \tag{66}$$

are already pairwise distinct. We then discard all integers $t \leq K$ from the set $S$, and by a slight abuse of notation, continue to denote the resulting subset by the same symbol $S$.

Finally, for each partition $\{U_p^t\}_{p=1}^{\alpha_t}$, we denote by $U^t(1)$ the unique set that contains the index 1.

*(i) The intersection of $K$ sets $U^1(1), U^2(1), \ldots, U^K(1)$ is precisely $\{1\}$, i.e.,*

$$U^1(1) \cap U^2(1) \cap \cdots \cap U^K(1) = \{1\}. \tag{67}$$

Indeed, since $1 \in U^t(1)$ for all $t = 1, \ldots, K$, it follows immediately that

$$1 \in U^1(1) \cap U^2(1) \cap \cdots \cap U^K(1). \tag{68}$$

Suppose, for the sake of contradiction, that there exists some $i \in [h]$ with $i > 1$ such that

$$i \in U^1(1) \cap U^2(1) \cap \cdots \cap U^K(1). \tag{69}$$

By the construction of $U^t(1)$, this assumption implies that $A_1^{k,t} = A_i^{k,t}$ for all $t = 1, \ldots, K$. Equivalently, the infinite sequences $\{A_1^{k,n}\}_{n\geq 1}$ and $\{A_i^{k,n}\}_{n\geq 1}$ coincide. This, however, contradicts the fact that their finite truncations

$$\{A_1^{k,n}\}_{n=1}^K, \quad \{A_2^{k,n}\}_{n=1}^K, \quad \ldots, \quad \{A_h^{k,n}\}_{n=1}^K$$

are pairwise distinct by the choice of $K$.

Therefore, no such $i > 1$ can exist. The only common element across all $U^1(1), \ldots, U^K(1)$ is the index 1, which establishes the claim.

*(ii) For each $t = 1, \ldots, K$, define the set*

$$V^t := U^t(1) \cap \{1, 2, \ldots, m\} \subset \{1, 2, \ldots, m\}. \tag{70}$$

*Then, one has*

$$V^1 \cap V^2 \cap \cdots \cap V^K = \{1\}. \tag{71}$$

Indeed, one computes

$$\begin{aligned}
V^1 \cap V^2 \cap \cdots \cap V^K &= \bigcap_{t=1}^K \left( U^t(1) \cap \{1, \ldots, m\} \right) \\
&= \bigcap_{t=1}^K U^t(1) \cap \{1, \ldots, m\} \\
&= \{1\} \cap \{1, \ldots, m\} \\
&= \{1\}. \tag{72}
\end{aligned}$$

*(iii) Among the $K$ sets $V^1, \ldots, V^K$, there exists a positive integer $\gamma < m$ such that one can select $\gamma$ sets, say $V^{t_1}, \ldots, V^{t_\gamma}$ with $1 \leq t_1 < t_2 < \cdots < t_\gamma \leq K$, satisfying the following property: the intersection of these $\gamma$ sets is $\{1\}$, whereas the intersection of any $\gamma - 1$ among them is no longer $\{1\}$.*

To prove this, let $\gamma$ be the smallest positive integer such that there exist $\gamma$ sets among $V^1, \ldots, V^K$ whose intersection equals $\{1\}$. The existence of such a $\gamma$ is guaranteed since the intersection of all

$K$ sets is $\{1\}$. Denote these $\gamma$ sets by $V^{t_1}, \ldots, V^{t_\gamma}$. By the minimality of $\gamma$, if one removes any single set from $\{V^{t_1}, \ldots, V^{t_\gamma}\}$, the intersection of the remaining $\gamma - 1$ sets cannot be $\{1\}$.

It remains to show that $\gamma < m$. By minimality, it suffices to establish the existence of fewer than $m$ sets among $\{V^1, \ldots, V^K\}$ whose intersection is $\{1\}$. Since

$$V^1 \cap V^2 \cap \cdots \cap V^K = \{1\}, \tag{73}$$

for each $i \in \{2, \ldots, m\}$ there must exist at least one set among $V^1, \ldots, V^K$ that does not contain $i$. As there are $m - 1$ such indices $i$, we can collect at most $m - 1$ sets that collectively exclude all of these elements. Consequently, the intersection of these at most $m - 1$ sets is $\{1\}$, which proves $\gamma \leq m - 1 < m$.

This completes the proof. The argument is essentially a pigeonhole-type principle: since every element $i \in \{2, \ldots, m\}$ must be excluded by at least one set, and there are $m - 1$ such elements in total, at most $m - 1$ sets suffice to ensure that all of them are removed, leaving only 1 in the intersection.

*(iv) In those $\gamma$ sets $V^{t_1}, \ldots, V^{t_\gamma}$ in (iii), for each $i \in [\gamma]$, one can choose $v_i \in V^{t_i}$ such that $v_1, \ldots, v_\gamma$ are pairwise distinct.*

This is a standard application of the Hall Marriage Theorem (see Appendix B.3.2). For convenience, rename $V^{t_i}$ as $W^i$ for $i \in [\gamma]$. For each $k \in \{1, \ldots, \gamma\}$, by assumption, we may choose

$$b_k \in \left( \bigcap_{i \neq k} W^i \right) \setminus \{1\}. \tag{74}$$

By construction, $b_k \neq 1$, and $b_k \in W^i$ for all $i \neq k$. Moreover, $b_k \notin W^k$, since otherwise $b_k$ would belong to $\bigcap_{i=1}^\gamma W^i = \{1\}$, a contradiction. Let $B = \{b_1, \ldots, b_\gamma\}$. Consider the bipartite graph with left vertices $\{W^1, \ldots, W^\gamma\}$ and right vertices $\{1\} \cup B \subseteq \{1, \ldots, m\}$, with an edge $W^i \leftrightarrow x$ whenever $x \in W^i$. A system of distinct representatives (SDR) of size $\gamma$ in this graph yields the desired elements $v_i \in W^i$. By Hall's theorem, it suffices to show that for every nonempty $J \subseteq \{1, \ldots, \gamma\}$, the neighborhood $N(J)$ satisfies $|N(J)| \geq |J|$.

- If $|J| = 1$, say $J = \{i\}$, then $1 \in W^i$. Furthermore, for every $k \neq i$ we have $b_k \in W^i$. Thus

$$|N(J)| \geq 1 + (\gamma - 1) = \gamma \geq |J|. \tag{75}$$

- If $|J| \geq 2$, fix $k \in \{1, \ldots, \gamma\}$.
    - If $k \notin J$, then $b_k \in W^i$ for every $i \in J$, hence $b_k \in N(J)$.
    - If $k \in J$, pick any $j \in J \setminus \{k\}$. Since $b_k \in W^j$, it follows that $b_k \in N(J)$.

  Thus every $b_k$ belongs to $N(J)$, and clearly $1 \in N(J)$. Hence

$$|N(J)| \geq |B| + 1 = \gamma + 1 \geq |J|. \tag{76}$$

Since Hall's condition is satisfied, there exists a matching that assigns to each $W^i$ a distinct element of $\{1\} \cup B$ contained in $W^i$. These assigned elements provide the required representatives $v_i \in W^i$, which are pairwise distinct.

**Step 5.**

To deliver the result of this part, we now employ the token indices $t_1, \ldots, t_\gamma$ identified in *(iii)* and *(iv)* of **Step 4**, together with the token indices in the set $S$ also obtained in **Step 4**. We recall the properties of these token indices that will be used:

1. For all $t \in S$, the partition $\{U_p^t\}_{p=1}^{\alpha_t}$, defined in **Step 4**, coincides with $\{U_p\}_{p=1}^\alpha$. In particular, by reindexing the head indices, we may assume $U_1 = \{1, \ldots, m\}$. This guarantees that the structure of the partition is stable across infinitely many $t \in S$, providing us with a consistent reference framework.

2. For all $t_i$ with $i \in [\gamma]$, where $\gamma < m$, recall that $V^{t_i} = U^{t_i}(1) \cap \{1, \ldots, m\}$. One can select $\gamma$ head indices $v_i \in V^{t_i}$ such that they are pairwise distinct. This property will be crucial later when we need to ensure that certain representatives can be chosen without overlap.

We also recall the main result from **Step 3**, namely Equation (62): for any $(s_1, \ldots, s_h) \in [L]^h$ with pairwise distinct entries, and for each $i \in [h]$, one has

$$0 = \sum_{\sigma \in S_h \,:\, A_j^{k,s_j} = A_{\sigma(j)}^{k,s_j} \; \forall j \in [h]} B_{\sigma(i)}. \tag{77}$$

This identity is the foundation of the argument: it asserts that, under the given matching condition on the coefficients $A_j^{k,s_j}$, a nontrivial linear combination of the $B_i$'s must vanish.

Now, in Equation (77), let us consider $(s_1, \ldots, s_h) \in [L]^h$ constructed as follows. First, observe that the index set $\{1, \ldots, h\}$ can be decomposed into three disjoint parts:

$$\{1, \ldots, h\} = \{v_1, \ldots, v_\gamma\} \sqcup \big(\{1, \ldots, m\} \setminus \{v_1, \ldots, v_\gamma\}\big) \sqcup \big(U_2 \sqcup U_3 \sqcup \cdots \sqcup U_\alpha\big). \tag{78}$$

The first component corresponds to the specially chosen distinct representatives $v_i$, the second to the remaining elements of $U_1$, and the third to all indices belonging to the other partition classes $U_2, \ldots, U_\alpha$.

Now fix a subset $T \subset [\gamma]$. Define $(s_1, \ldots, s_h) \in [L]^h$ by setting, for each $j \in [h]$,

1. If $j = v_i$ for some $i \in T$, then set $s_j = s_{v_i} = t_i$. In other words, the positions corresponding to $T$ are aligned with the distinguished token indices $t_i$.

2. If $j \in \{1, \ldots, m\} \setminus \{v_i : i \in T\}$, take $s_j$ to be an arbitrary element of $S$. This ensures consistency with the partition structure while leaving us flexibility in the assignment.

3. If $j \in U_p$ for some $2 \le p \le \alpha$, then take $s_j$ to be an arbitrary element of $S$. Again, this choice respects the partitioning of indices into classes $U_p$.

For the chosen $(s_1, \ldots, s_h) \in [L]^h$, we analyze which $\sigma \in S_h$ satisfy the condition $A_j^{k,s_j} = A_{\sigma(j)}^{k,s_j}$ for all $j \in [h]$. We make the following observations, case by case:

1. For $j \in U_2 \sqcup U_3 \sqcup \cdots \sqcup U_\alpha$, say $j \in U_p$ with $2 \le p \le \alpha$, the condition $A_j^{k,s_j} = A_{\sigma(j)}^{k,s_j}$ implies $\sigma(j) \in U_p$. Hence

$$\sigma(U_2 \sqcup U_3 \sqcup \cdots \sqcup U_\alpha) = U_2 \sqcup U_3 \sqcup \cdots \sqcup U_\alpha, \tag{79}$$

and consequently $\sigma(U_1) = U_1$. In particular, if $j \in U_1$, then $\sigma(j) \in U_1$.

2. For $j \in \{1, \ldots, m\} \setminus \{v_i : i \in T\}$, if $A_j^{k,s_j} = A_{\sigma(j)}^{k,s_j}$, then necessarily $\sigma(j) \in U_1 = \{1, \ldots, m\}$. Thus the entire set $U_1$ is stable under $\sigma$, but the specific images of these indices may vary within $U_1$.

3. For $j = v_i$ with $i \in T$, if $A_j^{k,s_j} = A_{\sigma(j)}^{k,s_j}$, then $\sigma(j) \in U^{s_{v_i}}(1) = U^{t_i}(1)$. From the previous point, we also know $\sigma(j) \in U_1$. Taken together, these conditions imply that $\sigma(j) \in V^{t_i} = U^{t_i}(1) \cap U_1$. In other words, the image of $v_i$ under $\sigma$ is constrained to lie inside the restricted set $V^{t_i}$.

Therefore, specifying a $\sigma \in S_h$ that satisfies $A_j^{k,s_j} = A_{\sigma(j)}^{k,s_j}$ for all $j \in [h]$ is equivalent to:

1. For each $j = v_i$ with $i \in T$, choosing $\sigma(j) = \sigma(v_i) \in V^{t_i}$,

2. For each $j \in \{1, \ldots, m\} \setminus \{v_i : i \in T\}$, choosing $\sigma(j) \in U_1 \setminus \{\sigma(v_i) : i \in T\}$ arbitrarily,

3. For each $j \in U_p$ with $2 \le p \le \alpha$, choosing $\sigma(j) \in U_p$.

In conclusion, the structure of admissible permutations $\sigma$ in Equation (77) is fully determined by the subset $T \subset [\gamma]$ and the representatives $v_i \in V^{t_i}$ chosen in **Step 4**. This description clarifies how the constraints arising from the partition classes $U_p$ and the distinguished representatives $v_i$ together restrict the allowed form of $\sigma$. Consequently, the sum in Equation (77) can be partitioned into contributions indexed by subsets $T \subset [\gamma]$, which will be the key mechanism for deriving vanishing conditions on the $B_i$'s in the subsequent step.

With these observations in hand, we now perform explicit computations. Fix one choice of $(s_1, \ldots, s_h) \in [L]^h$ satisfying the above construction, and in Equation (77) take $i = v_i$ for some $i \in T$. The equation then specializes to

$$0 = \sum_{\sigma \in S_h \,:\, A_j^{k,t_j} = A_{\sigma(j)}^{k,t_j} \; \forall j \in [h]} B_{\sigma(v_i)}$$

$$= \sum_{v \in V^{t_i}} B_v \cdot \Big(\text{the number of } h\text{-tuples in the Cartesian product}$$

$$\prod_{j \in T} V^{t_j} \times U_1^{m-|T|} \times \prod_{p=2}^{\alpha} U_p^{|U_p|},$$

$$\text{such that all } h \text{ entries are pairwise distinct, and}$$

$$\text{the coordinate corresponding to } V^{t_i} \text{ is fixed to be } v\Big). \tag{80}$$

The interpretation is as follows: each valid permutation $\sigma$ contributes one admissible tuple, and the contribution is grouped according to which element $v \in V^{t_i}$ is assigned to the coordinate corresponding to $V^{t_i}$. The factor multiplying $B_v$ therefore counts exactly the number of such admissible tuples.

Now, observe that once the coordinates corresponding to the $V^{t_j}$'s are chosen, all the remaining coordinates can be filled freely within their respective partition blocks. In particular:

- The indices in $\{1, \ldots, m\} \setminus \{v_i : i \in T\}$ may be permuted arbitrarily within $U_1$, yielding a factor of $(m - |T|)!$.

- For each $p \in \{2, \ldots, \alpha\}$, the indices in $U_p$ may also be permuted arbitrarily, contributing a factor of $|U_p|!$.

Hence the above expression simplifies to

$$0 = \sum_{v \in V^{t_i}} B_v \cdot (m - |T|)! \cdot \prod_{p=2}^{\alpha} |U_p|!$$

$$\cdot \Big(\text{the number of } h\text{-tuples in the Cartesian product} \prod_{j \in T} V^{t_j},$$

$$\text{such that all entries are pairwise distinct, and}$$

$$\text{the coordinate corresponding to } V^{t_i} \text{ equals } v\Big). \tag{81}$$

Since the factorial factors are nonzero constants independent of the choice of $v$, we may divide them out to obtain the equivalent condition

$$0 = \sum_{v \in V^{t_i}} B_v \cdot \Big(\text{the number of } h\text{-tuples in the Cartesian product} \prod_{j \in T} V^{t_j},$$

$$\text{such that all entries are pairwise distinct, and}$$

$$\text{the coordinate corresponding to } V^{t_i} \text{ equals } v\Big). \tag{82}$$

This identity holds for every choice of subset $T \subset [\gamma]$ and for every $v \in V^{t_i}$ with $i \in [\gamma]$. The key point is that the coefficients $B_v$ appear only through such linear relations, weighted by combinatorial counts of admissible tuples. By applying Corollary B.10, we deduce that

$$0 = \sum_{i \in V^{t_1} \cap V^{t_2} \cap \cdots \cap V^{t_\gamma}} B_i. \tag{83}$$

Finally, recall from the construction in *(iii)* of **Step 4** that the intersection $V^{t_1} \cap V^{t_2} \cap \cdots \cap V^{t_\gamma}$ is exactly $\{1\}$. Therefore, the above equation reduces to

$$B_1 = 0, \tag{84}$$

We have established that $B_1 = 0$. By the preceding argument at the beginning of the proof, this immediately implies that all $B_i$ vanish identically. Hence, we conclude that $B_i = 0$ for every $i$, which completes the proof. $\qquad\square$

We have the following corollary of Theorem B.1.

**Corollary B.2.** *Consider two* MHA *maps with $h$ and $\bar{h}$ heads, parameterized by $\theta$ and $\bar{\theta}$, respectively. Assume that $A_i^{m,n}$ and $\bar{A}_i^{m,n}$ are nonzero for all feasible triples $(i, m, n)$. If the two* MHA *maps are identical, i.e.,*

$$\text{MHA}(\mathbf{x}; \theta) = \text{MHA}(\mathbf{x}; \bar{\theta}), \tag{85}$$

*then for every parameter family*

$$\{A^{m,n}\}_{m,n \geq 1} \subset \mathbb{R}^{d \times d}, \tag{86}$$

*we have the identity*

$$\sum_{i \in [h]\,:\,\{A_i^{m,n}\}_{m,n} = \{A^{m,n}\}_{m,n}} B_i = \sum_{i \in [\bar{h}]\,:\,\{\bar{A}_i^{m,n}\}_{m,n} = \{A^{m,n}\}_{m,n}} \bar{B}_i. \tag{87}$$

*Proof.* This follows directly from Theorem B.1. $\qquad\square$

### B.3 KEY LEMMAS FOR THE FUNCTIONAL EQUIVALENCE OF GENERAL MULTIHEAD ATTENTION

In this section, we introduce the preliminary concepts and fundamental results that will serve as the foundation for the proofs of our main theorems.

#### B.3.1 A RESULT ON THE LINEAR INDEPENDENCE OF EXPONENTIAL POLYNOMIALS OVER THE FIELD OF RATIONAL FUNCTIONS

Let $n$ be a positive integer. Recall that $\mathbb{R}[\mathbf{x}] = \mathbb{R}[x_1, \ldots, x_n]$ denotes the polynomial ring in $n$ variables over $\mathbb{R}$. Its field of fractions is denoted by $\mathbb{R}(\mathbf{x})$, that is,

$$\mathbb{R}(\mathbf{x}) = \left\{ \frac{p}{q} \,:\, p, q \in \mathbb{R}[\mathbf{x}],\ q \neq 0 \right\}, \tag{88}$$

the field of all rational functions in the variables $x_1, \ldots, x_n$ with real coefficients.

We now state and prove a standard result concerning the linear independence of exponential polynomials over $\mathbb{R}(\mathbf{x})$.

**Lemma B.3.** *Let $p_1, \ldots, p_m$ be polynomials in $\mathbb{R}[\mathbf{x}]$ such that $p_i - p_j$ is nonconstant whenever $i \neq j$. Suppose $q_1, \ldots, q_m$ are rational functions in $\mathbb{R}(\mathbf{x})$ satisfying*

$$q_1 \cdot e^{p_1} + \cdots + q_m \cdot e^{p_m} = 0. \tag{89}$$

*Then necessarily $q_1 = \cdots = q_m = 0$.*

*Proof.* We proceed by induction on $m$.

*Base case.*

For $m = 1$, the statement is immediate. Indeed, if $q_1 \cdot e^{p_1} = 0$, then since $e^{p_1}$ never vanishes, it follows that $q_1 = 0$.

*Inductive step.*

Assume the result holds for every collection of fewer than $m$ exponentials. Let $q_1, \ldots, q_m \in \mathbb{R}(\mathbf{x})$ satisfy

$$q_1 \cdot e^{p_1} + \cdots + q_m \cdot e^{p_m} = 0. \tag{90}$$

We wish to show that all $q_i$ vanish. Suppose, for contradiction, that not all $q_i$ are zero. Without loss of generality, assume $q_m \neq 0$.

Dividing through Equation (90) by $q_m e^{p_m}$ yields

$$\frac{q_1}{q_m} \cdot e^{p_1 - p_m} + \cdots + \frac{q_{m-1}}{q_m} \cdot e^{p_{m-1} - p_m} + 1 = 0. \tag{91}$$

This expresses 1 as a linear combination of the exponentials $e^{p_j - p_m}$ with coefficients in $\mathbb{R}(\mathbf{x})$.

Now differentiate both sides of Equation (91) with respect to each variable $x_i$ for $i = 1, \ldots, n$. Since the derivative of 1 is zero, we obtain

$$\sum_{j=1}^{m-1} \left( \frac{\partial}{\partial x_i} \left( \frac{q_j}{q_m} \right) + \frac{q_j}{q_m} \cdot \frac{\partial}{\partial x_i} (p_j - p_m) \right) e^{p_j - p_m} = 0. \tag{92}$$

Each coefficient in parentheses lies in $\mathbb{R}(\mathbf{x})$.

Since $p_1 - p_m, \ldots, p_{m-1} - p_m$ are pairwise distinct and nonconstant, the corresponding exponentials $e^{p_j - p_m}$ are linearly independent over $\mathbb{R}(\mathbf{x})$ by the induction hypothesis. Therefore, each coefficient in Equation (92) must vanish, i.e.,

$$\frac{\partial}{\partial x_i} \left( \frac{q_j}{q_m} \right) + \frac{q_j}{q_m} \cdot \frac{\partial}{\partial x_i} (p_j - p_m) = 0, \tag{93}$$

for every $i = 1, \ldots, n$ and $j = 1, \ldots, m - 1$. Equivalently,

$$\frac{\partial}{\partial x_i} \left( \frac{q_j}{q_m} \cdot e^{p_j - p_m} \right) = 0. \tag{94}$$

This shows that for each $j = 1, \ldots, m - 1$, the function

$$\frac{q_j}{q_m} \cdot e^{p_j - p_m} \tag{95}$$

is independent of all variables $x_1, \ldots, x_n$, and hence must be a constant $c_j \in \mathbb{R}$.

If some $c_j \neq 0$, then $q_j \neq 0$ and we would have

$$e^{p_j - p_m} = \frac{c_j q_m}{q_j}, \tag{96}$$

which would imply that $e^{p_j - p_m}$ is a rational function, and therefore constant. This contradicts the assumption that $p_j - p_m$ is nonconstant.

Thus, each $c_j = 0$, forcing $q_j = 0$ for all $j = 1, \ldots, m - 1$. Substituting into Equation (91) then yields $1 = 0$, an impossibility.

Hence our assumption was false, and all $q_i = 0$. By induction, the lemma follows. $\qquad \square$

Lemma B.3 is crucial for arguments in Theorem B.1, involving exponential polynomials over $\mathbb{R}(\mathbf{x})$.

### B.3.2 HALL'S MARRIAGE THEOREM AND SYSTEMS OF DISTINCT REPRESENTATIVES

In this section, we recall a classical result from combinatorics, known as *Hall's Marriage Theorem* (Hall, 1935), which provides necessary and sufficient conditions for the existence of a system of distinct representatives (SDR). This theorem will play a crucial role in our arguments, as our construction ultimately reduces to the problem of selecting distinct representatives from a family of subsets. Let $\mathcal{A} = \{A_1, A_2, \ldots, A_s\}$ be a finite family of subsets of a ground set $X$. A *system of distinct representatives* (SDR) for $\mathcal{A}$ is a set $\{a_1, a_2, \ldots, a_s\}$ such that $a_i \in A_i$ for each $i$ and all $a_1, \ldots, a_s$ are pairwise distinct. Equivalently, an SDR is an injective choice function assigning to each $A_i$ an element $a_i \in A_i$.

The existence of an SDR is a classical question in combinatorics, and Hall's theorem provides a complete characterization.

**Theorem B.4** (Hall's Marriage Theorem). *Let $\mathcal{A} = \{A_1, A_2, \ldots, A_s\}$ be a finite family of subsets of a set $X$. Then $\mathcal{A}$ admits a system of distinct representatives if and only if the following condition (Hall's condition) holds:*

$$\left| \bigcup_{i \in J} A_i \right| \geq |J| \quad \text{for every subset } J \subseteq \{1, 2, \ldots, s\}. \tag{97}$$

**Remark B.5.** In words, Hall's condition states that for every subcollection of the sets $A_i$, the total number of available elements in their union must be at least as large as the number of sets in the subcollection. This condition is clearly necessary: if $|J|$ sets are assigned representatives, then at least $|J|$ distinct elements are required. The theorem asserts that this necessary condition is also sufficient.

Hall's Marriage Theorem plays a central role in the argument of Theorem B.1. Moreover, its application is closely connected to the statements of Theorem B.8 and Corollary B.10.

### B.3.3 THE MÖBIUS FUNCTION ON THE PARTITION LATTICE

This section introduces the necessary background on incidence algebras and Möbius inversion over finite posets. We then establish an identity for the Möbius function that will serve as a fundamental tool throughout the remainder of the paper. We also present several connections between this identity and other well-studied combinatorial concepts, with the aim of providing readers with greater intuition about its significance. For comprehensive treatments of these topics, we refer the reader to (Rota, 1964; Stanley, 2011).

#### Incidence Algebras and Möbius Inversion on Finite Posets

Let $(P, \leq)$ be a finite poset. The *incidence algebra* $I(P)$ over $\mathbb{C}$ consists of all functions

$$f := \{(x, y) \in P \times P : x \leq y\} \longrightarrow \mathbb{C}. \tag{98}$$

with convolution

$$(f * g)(x; y) := \sum_{x \leq z \leq y} f(x; z)\, g(z; y), \quad \text{for all } x \leq y. \tag{99}$$

The identity for convolution is the Kronecker delta $\delta(x, y)$ (i.e. $\delta(x, y) = 1$ if $x = y$, and 0 otherwise). The *zeta function* $\zeta \in I(P)$ is $\zeta(x, y) \equiv 1$ for $x \leq y$. An element $f \in I(P)$ is invertible if and only if $f(x, x) \neq 0$ for all $x \in P$; in that case $f^{-1}$ is its inverse under convolution.

**Möbius function.** The *Möbius function* $\mu = \mu_P \in I(P)$ is defined as the convolution inverse of $\zeta$:

$$\mu * \zeta = \zeta * \mu = \delta. \tag{100}$$

Equivalently, for all $x \leq y$ in $P$, one has

$$\sum_{x \leq z \leq y} \mu(x; z) = \delta(x; y). \tag{101}$$

As a consequence, if $f, g : P \to \mathbb{C}$ satisfy

$$f(x) = \sum_{y \geq x} g(y), \quad \text{for all } x \in P, \tag{102}$$

then *Möbius inversion* yields

$$g(x) = \sum_{y \geq x} \mu(x; y)\, f(y), \quad \text{for all } x \in P. \tag{103}$$

**Products of posets.** If $P, Q$ are finite posets, their product $P \times Q$ is ordered componentwise. Define

$$(\zeta_P \otimes \zeta_Q)\big((p_1, q_1); (p_2, q_2)\big) := \zeta_P(p_1; p_2)\, \zeta_Q(q_1; q_2). \tag{104}$$

A direct computation in $I(P \times Q)$ shows

$$\zeta_{P \times Q} = \zeta_P \otimes \zeta_Q, \tag{105}$$

$$(\mu_P \otimes \mu_Q) * (\zeta_P \otimes \zeta_Q) = \delta_P \otimes \delta_Q = \delta_{P \times Q}. \tag{106}$$

Hence

$$\mu_{P \times Q}\big((p_1, q_1); (p_2, q_2)\big) = \mu_P(p_1; p_2)\, \mu_Q(q_1; q_2). \tag{107}$$

**The Partition Lattice and Interval Factorization** Let $U$ be a finite set with $|U| = n$. The set $\Pi(U)$ of all set partitions of $U$, ordered by refinement, forms a finite lattice with minimum $\hat{0}$ (all singletons) and maximum $\hat{1}$ (one block). The goal of this section is to derive the following explicit formula, stated in the following theorem:

**Theorem B.6.** *For $\pi \in \Pi(U)$, one has:*

$$\mu_{\Pi(U)}(\hat{0}, \pi) = \prod_{B \in \pi} (-1)^{|B|-1}(|B| - 1)!. \tag{108}$$

For clarity, we begin with an outline of the proof. The reasoning unfolds in two stages.

1. **Interval factorization.** Restriction to blocks induces a canonical isomorphism:

$$[\hat{0}, \pi] \cong \prod_{B \in \pi} \Pi(B). \tag{109}$$

By multiplicativity of the Möbius function on products, one has:

$$\mu_{\Pi(U)}(\hat{0}, \pi) = \prod_{B \in \pi} \mu_{\Pi(B)}(\hat{0}_B, \hat{1}_B). \tag{110}$$

2. **One–block evaluation.** Using the exponential formula for labelled set partitions, for all $n \geq 1$, one has:

$$\mu_{\Pi([n])}(\hat{0}, \hat{1}) = (-1)^{n-1}(n - 1)!. \tag{111}$$

Substituting into the product from Step 1 yields

$$\mu_{\Pi(U)}(\hat{0}, \pi) = \prod_{B \in \pi} (-1)^{|B|-1}(|B| - 1)!. \tag{112}$$

Having outlined the strategy, we now provide the full proof with all intermediate steps made explicit.

*Proof.* We structure the proof into several steps for the sake of clarity and readability.

**Step 1 (Interval factorization in the partition lattice).**

A partition $\pi \in \Pi(U)$ is a set of disjoint nonempty blocks $B \subseteq U$ covering $U$. For $\sigma, \pi \in \Pi(U)$ write $\sigma \leq \pi$ if every block of $\sigma$ is contained in a block of $\pi$. For $\sigma \leq \pi$ and a block $B \in \pi$, let $\sigma|_B$ be the restriction of $\sigma$ to $B$ (intersect each block of $\sigma$ with $B$ and remove empties). Denote by $\hat{1}_B$ the one-block partition of $B$. We have the following result.

**Lemma B.7** (Interval factorization). *For $\sigma \leq \pi$ in $\Pi(U)$, restriction induces a poset isomorphism*

$$\Phi : [\sigma, \pi] \longrightarrow \prod_{B \in \pi} \Pi\big(\sigma|_B; \hat{1}_B\big), \qquad \Phi(\tau) : \big(\tau|_B\big)_{B \in \pi}. \tag{113}$$

*Its inverse maps $(\rho_B)_{B \in \pi}$ to the join $\bigvee_{B \in \pi} \rho_B$, which coincides with the partition whose restriction to each $B$ equals $\rho_B$.*

*Proof.* If $\tau \in [\sigma, \pi]$, then $\sigma \leq \tau \leq \pi$ implies that each block of $\tau$ lies inside some block of $\pi$, so $\tau|_B$ is a partition of $B$ refining $\sigma|_B$, hence $\sigma|_B \leq \tau|_B \leq \hat{1}_B$. Thus $\Phi$ is well-defined and order-preserving. Conversely, if $(\rho_B)_{B \in \pi}$ satisfies $\sigma|_B \leq \rho_B \leq \hat{1}_B$, define $\rho$ by declaring that $x, y \in U$ lie in the same block of $\rho$ iff either $x, y \in B$ and $x \sim_{\rho_B} y$ for some $B \in \pi$, or $x, y$ lie in different blocks of $\pi$ (which never happens since we work blockwise). Then $\rho$ is a partition with $\sigma \leq \rho \leq \pi$ and $\rho|_B = \rho_B$. One checks $\Phi(\rho) = (\rho_B)$ and $\bigvee_{B \in \pi}(\tau|_B) = \tau$, hence $\Phi$ is an isomorphism. $\square$

Setting $\sigma = \hat{0}$ in Lemma B.7 yields

$$[\hat{0}, \pi] \cong \prod_{B \in \pi} \Pi(B). \tag{114}$$

Applying the multiplicativity Equation (107) to Equation (114), one has

$$\mu_{\Pi(U)}(\hat{0}, \pi) = \prod_{B \in \pi} \mu_{\Pi(B)}(\hat{0}_B, \hat{1}_B). \tag{115}$$

Therefore, to compute $\mu_{\Pi(U)}(\hat{0}, \pi)$ for arbitrary $\pi$, it suffices to evaluate the single-block quantity

$$m(n) \coloneqq \mu_{\Pi_n}(\hat{0}; \hat{1}), \tag{116}$$

for $n \in \mathbb{N}$, where $\Pi_n$ denotes the partition lattice on an $n$-element set.

**Step 2 (The one-block value via the exponential formula for labeled set partitions).**

We now determine $m(n)$ exactly. One has a Möbius sum constraint as follows: by Equation (101), for every finite poset and any $x < y$, one has

$$\sum_{x \leq z \leq y} \mu(x; z) = 0. \tag{117}$$

In $\Pi_n$, taking $x = \hat{0}$ and $y = \hat{1}$ gives

$$\sum_{\tau \in \Pi_n} \mu_{\Pi_n}(\hat{0}, \tau) = 0, \tag{118}$$

for all $n \geq 2$. For $n = 0, 1$, the sum equals 1 (the unique element of the interval). By Equation (115) applied inside $\Pi_n$, one has

$$\mu_{\Pi_n}(\hat{0}; \tau) = \prod_{B \in \tau} m(|B|). \tag{119}$$

Define

$$F_n \coloneqq \sum_{\tau \in \Pi_n} \prod_{B \in \tau} m(|B|). \tag{120}$$

Then, for $n \geq 2$, one has

$$F_0 = 1, \qquad F_1 = 1, \qquad F_n = 0. \tag{121}$$

A standard labeled-partition identity (the exponential formula) asserts that for any sequence $(a_k)_{k \geq 1}$,

$$\sum_{n \geq 0} \left( \sum_{\tau \in \Pi_n} \prod_{B \in \tau} a_{|B|} \right) \frac{z^n}{n!} = \exp\left( \sum_{k \geq 1} a_k \frac{z^k}{k!} \right). \tag{122}$$

Applying this with $a_k = m(k)$ yields

$$\sum_{n \geq 0} F_n \frac{z^n}{n!} = \exp\left( \sum_{k \geq 1} m(k) \frac{z^k}{k!} \right). \tag{123}$$

Using Equation (121), the left-hand side of Equation (123) equals $1 + z$. Taking the formal logarithm gives

$$\sum_{k \geq 1} m(k) \frac{z^k}{k!} = \log(1 + z) = \sum_{k \geq 1} (-1)^{k-1} \frac{z^k}{k}. \tag{124}$$

Equating coefficients, for $k \geq 1$, one has

$$m(k) = k! \cdot \frac{(-1)^{k-1}}{k} = (-1)^{k-1} (k-1)!. \tag{125}$$

Substituting Equation (125) into the block factorization Equation (115) gives the desired expression in Equation (108):

$$\mu_{\Pi(U)}(\hat{0}, \pi) = \prod_{B \in \pi} (-1)^{|B|-1} (|B| - 1)!. \tag{126}$$

This concludes the proof. $\qquad\square$

The identity established in Theorem B.6 plays a pivotal role in the proof of Theorem B.8, which, in turn, functions as a supporting lemma for the proof of Theorem B.1–the main result of this work.

### B.4 A Technical Result on Weighted Sums over Distinct Tuples

We now present a result concerning the problem of weighted sums over distinct tuples. The results developed in this section form the backbone of our argument in the proof of Theorem B.1, the main result of this work.

**Theorem B.8.** *Given positive integers $m, n \geq 1$. For each $i \in [m]$, let $A_i$ be a subset of $[n]$. Let $x_1, \ldots, x_n$ be $n$ real numbers. For any nonempty $S \subseteq [m]$, define*

$$F_S := \Big\{ (a_i)_{i \in S} : a_i \in A_i \text{ for all } i \in S, \text{ and all } a_i\text{'s are pairwise distinct} \Big\}. \tag{127}$$

*For $i \in S$ and $a \in A_i$, define the fiber*

$$F_{S,i,a} := \{ (a_j)_{j \in S} \in F_S : a_i = a \}. \tag{128}$$

*For any nonempty $T \subseteq [m]$, define $A_T := \bigcap_{i \in T} A_i$, and*

$$G(T) := \sum_{a \in A_T} x_a. \tag{129}$$

*Assume that, for every nonempty $S \subseteq [m]$ and every $i \in S$, one has*

$$\sum_{a \in A_i} |F_{S,i,a}| \, x_a = 0. \tag{130}$$

*Then, for every nonempty $T \subseteq [m]$, one has*

$$G(T) = \sum_{a \in A_T} x_a = 0. \tag{131}$$

*Proof.* Let $S$ be a nonempty finite set. Denote by $\Pi(S)$ the lattice of set partitions of $S$ ordered by refinement: For $\sigma, \pi \in \Pi(S)$, we write $\sigma \leq \pi$ if every block of $\sigma$ is contained in a block of $\pi$. Any $\pi \in \Pi(S)$ is a family of disjoint nonempty blocks whose union is $S$. For a block $B \subseteq S$ define

$$A_B := \bigcap_{j \in B} A_j, \qquad \text{and} \qquad |A_B| := \Big| \bigcap_{j \in B} A_j \Big|. \tag{132}$$

Let $\mu$ denote the Möbius function of $\Pi(S)$ (with respect to refinement). $\mu$ is determined by $\sum_{\sigma: \sigma \leq \pi} \mu(\sigma) = \mathbf{1}_{\{\pi = \hat{0}\}}$, where $\hat{0}$ is the discrete partition. Formula (1) follows by multiplicativity of $\mu$ over blocks and the known one-block value $(-1)^{r-1}(r-1)!$ for a block of size $r$. It is well-known that:

$$\mu(\pi) = \prod_{B \in \pi} (-1)^{|B|-1} (|B| - 1)!. \tag{133}$$

Fix a nonempty $S \subseteq [m]$, an index $i \in S$, and an element $a \in [n]$. Let $\mathcal{G}_S$ be the set of all functions $g : S \to [n]$ satisfying $g(j) \in A_j$ for all $j \in S$ (note that, there is no distinctness condition). For $g \in \mathcal{G}_S$, define its equality partition $\pi(g) \in \Pi(S)$ by:

$$j \sim_{\pi(g)} k \quad \text{if and only if} \quad g(j) = g(k). \tag{134}$$

Thus $\pi(g)$ records which indices are assigned the same value by $g$. One has $g$ is injective on $S$ if and only if $\pi(g) = \hat{0}$. The set $F_S$ of injective choices can be described as:

$$F_S = \left\{ g \in \mathcal{G}_S : \ \pi(g) = \hat{0} \right\}, \tag{135}$$

and the *fiber* fixing the value at the distinguished index $i$ is:

$$F_{S,i,a} = \left\{ g \in \mathcal{G}_S : \ g(i) = a, \ \pi(g) = \hat{0} \right\}. \tag{136}$$

For $\pi \in \Pi(S)$ and $i \in S$, let $B_i(\pi)$ denote the unique block of $\pi$ containing $i$. Define:

$$N_{S,i,a}(\pi) := \left| \left\{ g \in \mathcal{G}_S : \ g \text{ is constant on each block of } \pi, \ g(i) = a \right\} \right|. \tag{137}$$

That is, $N_{S,i,a}(\pi)$ counts maps that are constant along blocks of $\pi$ (so the only equalities allowed among coordinates are those forced by $\pi$) and take the prescribed value $a$ at the index $i$. For every $\pi \in \Pi(S)$, one has:

$$N_{S,i,a}(\pi) = \mathbf{1}_{\{a \in A_{B_i(\pi)}\}} \prod_{\substack{B \in \pi \\ B \neq B_i(\pi)}} |A_B|. \tag{138}$$

Indeed, if $g$ is constant on each block of $\pi$, the value on the block $B_i(\pi)$ must equal $g(i) = a$. This is possible exactly when $a \in \bigcap_{j \in B_i(\pi)} A_j = A_{B_i(\pi)}$, which contributes the indicator $\mathbf{1}_{\{a \in A_{B_i(\pi)}\}}$. Then, for any other block $B \in \pi$ with $B \neq B_i(\pi)$, the common value of $g$ on $B$ can be chosen arbitrarily from the intersection $A_B = \bigcap_{j \in B} A_j$, independently across distinct blocks. Therefore there are $|A_B|$ choices for each such block, and multiplying over all $B \neq B_i(\pi)$ yields the product in Equation (138). Now, for $g \in \mathcal{G}_S$, define the two indicator functions on $\Pi(S)$:

$$E(g) := \mathbf{1}_{\{\pi(g) = \hat{0}\}}, \quad \text{and} \quad C_\pi(g) := \mathbf{1}_{\{\pi(g) \geq \pi\}} \quad (\pi \in \Pi(S)). \tag{139}$$

Here $\pi(g) \geq \pi$ means that $g$ is constant on every block of $\pi$. By general Möbius inversion on posets, one has:

$$E(g) \ = \ \sum_{\pi \in \Pi(S)} \mu(\pi) \, C_\pi(g), \tag{140}$$

since

$$\sum_{\sigma \leq \pi(g)} \mu(\sigma) = \mathbf{1}_{\{\pi(g) = \hat{0}\}}. \tag{141}$$

Now fix $i \in S$ and $a \in [n]$, multiply the last identity by $\mathbf{1}_{\{g(i) = a\}}$, and sum over all $g \in \mathcal{G}_S$, one has:

$$\left| F_{S,i,a} \right| = \sum_{g \in \mathcal{G}_S} \mathbf{1}_{\{g(i) = a\}} E(g) = \sum_{\pi \in \Pi(S)} \mu(\pi) \sum_{g \in \mathcal{G}_S} \mathbf{1}_{\{g(i) = a\}} C_\pi(g). \tag{142}$$

The inner sum is precisely $N_{S,i,a}(\pi)$ by definition. Using Equation (138), one therefore obtains the explicit expansion:

$$|F_{S,i,a}| = \sum_{\pi \in \Pi(S)} \mu(\pi) \, \mathbf{1}_{\{a \in A_{B_i(\pi)}\}} \prod_{\substack{B \in \pi \\ B \neq B_i(\pi)}} |A_B|. \tag{143}$$

Multiply Equation (143) by $x_a$ and sum over all $a \in A_i$ (equivalently, over all $a \in [n]$, since the indicator in Equation (143) already forces $a \in A_i$ when $i \in B_i(\pi)$):

$$\sum_{a \in A_i} |F_{S,i,a}| \, x_a = \sum_{\pi \in \Pi(S)} \mu(\pi) \left( \prod_{\substack{B \in \pi \\ B \neq B_i(\pi)}} |A_B| \right) \left( \sum_{a \in A_{B_i(\pi)}} x_a \right). \tag{144}$$

With the shorthand $G(T) := \sum_{a \in A_T} x_a$ this becomes

$$\sum_{a \in A_i} |F_{S,i,a}| x_a = \sum_{\pi \in \Pi(S)} \mu(\pi) \left( \prod_{\substack{B \in \pi \\ B \neq B_i(\pi)}} |A_B| \right) G\big(B_i(\pi)\big). \tag{145}$$

By the hypothesis, the left–hand side of Equation (145) is 0. Hence

$$0 = \sum_{\pi \in \Pi(S)} \mu(\pi) \left( \prod_{\substack{B \in \pi \\ B \neq B_i(\pi)}} |A_B| \right) G\big(B_i(\pi)\big), \tag{146}$$

for every nonempty $S \subseteq [m]$ and every $i \in S$. Observe that, in Equation (146), the term $G\big(B_i(\pi)\big)$ only involves nonempty subsets $B_i(\pi)$ with $i \in B_i(\pi) \subseteq S$.

Back to the problem. We now show that $G(T) = 0$ for every nonempty $T \subseteq [m]$ by induction on $k := |T|$. We use the Equation (133) and Equation (146) a lots.

*Base case.*

Let $T = \{i\}$ for some $i \in [m]$. Take $S = \{i\}$ in the given hypothesis, one has

$$\sum_{a \in A_i} |F_{S,i,a}| x_a = 0. \tag{147}$$

Since $S$ has one element, an injective choice on $S$ is just a choice of a value in $A_i$, hence $|F_{\{i\},i,a}| = \mathbf{1}_{\{a \in A_i\}}$. Therefore

$$0 = \sum_{a \in A_i} |F_{\{i\},i,a}| x_a = \sum_{a \in A_i} x_a = G(\{i\}), \tag{148}$$

which establishes the base case.

*Inductive step.*

Fix $k \geq 2$ and assume the claim holds for all nonempty $U \subseteq [m]$ with $|U| < k$, i.e., $G(U) = 0$ whenever $1 \leq |U| \leq k - 1$. Let $T \subseteq [m]$ with $|T| = k$, and fix any distinguished index $i \in T$. Apply Equation (146) with $S = T$, we analyze the sum over $\pi \in \Pi(T)$ by separating the one–block partition from the rest.

*(a) The contribution of the one–block partition.*

There is a unique partition $\pi^\star = \{T\}$ with a single block. For this partition we have $B_i(\pi^\star) = T$, and the product over $B \neq B_i(\pi^\star)$ is an empty product, hence equals 1 by convention. By Equation (133) with $|T| = k$, one has:

$$\mu(\pi^\star) = (-1)^{k-1}(k-1)!. \tag{149}$$

Thus, the term of Equation (146) corresponding to $\pi^\star$ equals

$$\mu(\pi^\star) \cdot 1 \cdot G\big(B_i(\pi^\star)\big) = (-1)^{k-1}(k-1)! G(T). \tag{150}$$

*(b) The contribution of all other partitions.*

Let $\pi \in \Pi(T)$ with $\pi \neq \pi^\star$. Then $B_i(\pi)$ is a proper, nonempty subset of $T$ (it still contains $i$ but does not equal $T$). Consequently $|B_i(\pi)| \leq k - 1$. By the inductive hypothesis,

$$G\big(B_i(\pi)\big) = 0.$$

Hence every summand in Equation (146) with $\pi \neq \pi^\star$ vanishes, regardless of the multiplicative factor $\prod_{B \neq B_i(\pi)} |A_B|$ and the value of $\mu(\pi)$.

Collecting (a) and (b), identity Equation (146) with $S = T$ reduces to

$$0 = (-1)^{k-1}(k-1)! G(T). \tag{151}$$

Since $(-1)^{k-1}(k-1)! \neq 0$, we conclude $G(T) = 0$.

By induction on $k$, the relation $G(T) = 0$ holds for every nonempty $T \subseteq [m]$. $\qquad \square$

**Remark B.9** (Combinatorial intuition). Viewed combinatorially, $F_S$ is precisely the set of systems of distinct representatives (SDRs) for the family $\{A_i : i \in S\}$ For a fixed index $i \in S$ and value $a \in A_i$, the fiber $F_{S,i,a}$ enumerates those SDRs that assign the representative $a$ to position $i$. Assumption in Equation (154) therefore states that the weighted sum $\sum_{a \in A_i} |F_{S,i,a}| \, x_a$ vanishes for every nonempty $S \subseteq [m]$ and every $i \in S$; equivalently, the vector $x = (x_a)_{a \in [n]}$ is orthogonal to the vector of SDR–completion counts at coordinate $i$. Applying Möbius inversion on the Boolean lattice $(2^{[m]}, \subseteq)$ transfers these linear relations, with coefficients given by SDR multiplicities, into relations with unit coefficients, thereby collapsing the fiber-weighted sums to the unweighted intersection sums $\sum_{a \in \cap_{j \in T} A_j} x_a$. This mirrors the classical rook-polynomial/inclusion-exclusion paradigm: counts of placements with multiplicities invert to simple intersection counts once the incidence algebra is diagonalized by the Möbius function.

We have a direct corollary of Theorem B.8.

**Corollary B.10.** *Given positive integers $m, n \geq 1$. For each $i \in [m]$, let $A_i$ be a subset of $[n]$. Let $x_1, \ldots, x_n$ be $n$ real numbers. For any nonempty $S \subseteq [m]$, define*

$$F_S := \left\{ (a_i)_{i \in S} : a_i \in A_i \text{ for all } i \in S, \text{ and all } a_i\text{'s are pairwise distinct} \right\}. \tag{152}$$

*For $i \in S$ and $a \in A_i$, define the fiber*

$$F_{S,i,a} := \{ (a_j)_{j \in S} \in F_S : a_i = a \}. \tag{153}$$

*Assume that, for every nonempty $S \subseteq [m]$ and every $i \in S$, one has*

$$\sum_{a \in A_i} |F_{S,i,a}| \, x_a = 0. \tag{154}$$

*Then, one has*

$$G(T) = \sum_{a \in A_1 \cap \ldots \cap A_m} x_a = 0. \tag{155}$$

*Proof.* By taking $T = [m]$ in Theorem B.8, one obtains the asserted main conclusion. □

### B.5 PROOF OF THEOREM 3.2

**Theorem B.11** (Theorem 3.2 in the main paper). *Let*

$$\theta = \left( W_i^Q, W_i^K, W_i^V, W_i^O \right)_{i=1}^{h} \in \Omega_h, \text{ and } \bar{\theta} = \left( \bar{W}_i^Q, \bar{W}_i^K, \bar{W}_i^V, \bar{W}_i^O \right)_{i=1}^{\bar{h}} \in \Omega_{\bar{h}}, \tag{156}$$

*be two parameterizations of $\mathrm{MHA}_{\mathrm{RoPE}}$ maps. Assume that:*

1. *In the first $\mathrm{MHA}_{\mathrm{RoPE}}$ map, for each head $i \in [h]$, the similarity score between two arbitrary tokens does not vanish, i.e.,*

$$W_i^Q (W_i^K)^\top + W_i^K (W_i^Q)^\top \text{ and } W_i^Q R^n (W_i^K)^\top, \tag{157}$$

   *for all non-zero integer $n$, are non-zero.*

2. *In the second $\mathrm{MHA}_{\mathrm{RoPE}}$ map, for each head $i \in [\bar{h}]$, the similarity score between two arbitrary tokens does not vanish, i.e.,*

$$\bar{W}_i^Q (\bar{W}_i^K)^\top + \bar{W}_i^K (\bar{W}_i^Q)^\top \text{ and } \bar{W}_i^Q R^n (\bar{W}_i^K)^\top, \tag{158}$$

   *for all non-zero integer $n$, are non-zero.*

3. *In the first $\mathrm{MHA}_{\mathrm{RoPE}}$ map, the similarity score maps are pairwise distinct, i.e.,*

$$\left\{ W_i^Q (W_i^K)^\top + W_i^K (W_i^Q)^\top, \{ W_i^Q R^n (W_i^K)^\top \}_{n \in \mathbb{Z}, n \neq 0} \right\}, \tag{159}$$

   *for $i = 1, \ldots, h$, are $h$ pairwise distinct families.*

4. *In the second* $\mathrm{MHA}_{\mathrm{RoPE}}$ *map, the similarity score maps are pairwise distinct, i.e.,*

$$\left\{ \bar{W}_i^Q (\bar{W}_i^K)^\top + \bar{W}_i^K (\bar{W}_i^Q)^\top, \{\bar{W}_i^Q R^n (\bar{W}_i^K)^\top\}_{n\in\mathbb{Z}, n\neq 0} \right\}, \tag{160}$$

*for* $i = 1, \ldots, \bar{h}$, *are* $h$ *pairwise distinct families.*

5. *In the first* $\mathrm{MHA}_{\mathrm{RoPE}}$ *map, all matrices* $W_i^Q, W_i^K, W_i^V, W_i^O$ *for* $i \in [h]$ *are of rank* $d_h$.

6. *In the second* $\mathrm{MHA}_{\mathrm{RoPE}}$ *map, all matrices* $\bar{W}_i^Q, \bar{W}_i^K, \bar{W}_i^V, W_i^O$ *for* $i \in [h]$ *are of rank* $d_h$.

*If the two* $\mathrm{MHA}_{\mathrm{RoPE}}$ *maps are identical, i.e.,*

$$\mathrm{MHA}_{\mathrm{RoPE}}(\cdot; \theta) = \mathrm{MHA}_{\mathrm{RoPE}}(\cdot; \bar{\theta}), \tag{161}$$

*then* $h = \bar{h}$, *and there exists a permutation* $\sigma \in S_h$ *and invertible matrices* $\{U_i\}_{i=1}^h \subset \mathrm{H}(d_h)$ *and* $\{V_i\}_{i=1}^h \subset \mathrm{GL}(d_h)$ *such that*

$$\begin{aligned}
\bar{W}_i^Q &= W_{\sigma(i)}^Q \cdot U_i^\top, \quad \bar{W}_i^K = W_{\sigma(i)}^K \cdot (U_i)^{-1}, \\
\bar{W}_i^V &= W_{\sigma(i)}^V \cdot V_i^\top, \quad \bar{W}_i^O = W_{\sigma(i)}^O \cdot (V_i)^{-1}.
\end{aligned} \tag{162}$$

*Proof.* For $i \in [h]$ and $m, n \geq 1$, denote

$$A_i^{m,n} := W_i^Q R^{m-n} (W_i^K)^\top, \text{ if } m \neq n \tag{163}$$

$$A_i^{m,m} := \frac{W_i^Q (W_i^K)^\top + W_i^K (W_i^Q)^\top}{2}, \text{ and} \tag{164}$$

$$B_i := W_i^V (W_i^O)^\top. \tag{165}$$

For $i \in [\bar{h}]$ and $m, n \geq 1$, denote

$$\bar{A}_i^{m,n} := \bar{W}_i^Q R^{m-n} (\bar{W}_i^K)^\top, \text{ if } m \neq n \tag{166}$$

$$\bar{A}_i^{m,m} := \frac{\bar{W}_i^Q (\bar{W}_i^K)^\top + \bar{W}_i^K (\bar{W}_i^Q)^\top}{2}, \text{ and} \tag{167}$$

$$\bar{B}_i := \bar{W}_i^V (\bar{W}_i^O)^\top. \tag{168}$$

Then, one has

$$\mathrm{MHA}\left(\mathbf{x} : \{\{A_i^{m,n}\}_{m,n}; B_i\}_{i=1}^h\right) = \mathrm{MHA}_{\mathrm{RoPE}}(\cdot; \theta), \tag{169}$$

and

$$\mathrm{MHA}\left(\mathbf{x} : \{\{\bar{A}_i^{m,n}\}_{m,n}; \bar{B}_i\}_{i=1}^{\bar{h}}\right) = \mathrm{MHA}_{\mathrm{RoPE}}(\cdot; \bar{\theta}). \tag{170}$$

Thus,

$$\mathrm{MHA}\left(\mathbf{x} : \{\{A_i^{m,n}\}_{m,n}; B_i\}_{i=1}^h\right) = \mathrm{MHA}\left(\mathbf{x} : \{\{\bar{A}_i^{m,n}\}_{m,n}; \bar{B}_i\}_{i=1}^{\bar{h}}\right). \tag{171}$$

From the condition 3, 4, the property of parameters from these maps fit to the setting of Corollary B.2, which is that $A_i^{m,n}$ and $\bar{A}_i^{m,n}$ are nonzero for all feasible triples $(i, m, n)$. Thus, for every parameter family

$$\{A^{m,n}\}_{m,n\geq 1} \subset \mathbb{R}^{d\times d}, \tag{172}$$

we have the identity

$$\sum_{i\in[h] \,:\, \{A_i^{m,n}\}_{m,n} = \{A^{m,n}\}_{m,n}} B_i = \sum_{i\in[\bar{h}] \,:\, \{\bar{A}_i^{m,n}\}_{m,n} = \{A^{m,n}\}_{m,n}} \bar{B}_i. \tag{173}$$

From condition 3, one has $h$ families of parameters

$$\{A_1^{m,n}\}_{m,n\geq 1}, \{A_2^{m,n}\}_{m,n\geq 1}, \ldots, \{A_h^{m,n}\}_{m,n\geq 1}, \tag{174}$$

are pairwise distinct. Together with Equation (173), consider

$$\{A^{m,n}\}_{m,n\geq 1} = \{A_i^{m,n}\}_{m,n\geq 1}, \tag{175}$$

one has the left-hand side of Equation (173) is equal to $B_i$. Thus,

$$B_i = \sum_{j\in[\bar{h}] \,:\, \{\bar{A}_j^{m,n}\}_{m,n\geq 1}=\{A_i^{m,n}\}_{m,n\geq 1}} \bar{B}_j. \tag{176}$$

Note that, since all the matrices $W_i^V$ and $W_i^O$ have rank $d_h$, it implies that all $B_i$ are non-zero. From Equation (176), for each $i \in [h]$, since the left-hand side is non-zero, the right-hand side has at least one index $j \in [\bar{h}]$ such that $\bar{B}_j$ is non-zero and $\{\bar{A}_j^{m,n}\}_{m,n\geq 1} = \{A_i^{m,n}\}_{m,n\geq 1}$. Since $h$ families of parameters

$$\{A_1^{m,n}\}_{m,n\geq 1}, \{A_2^{m,n}\}_{m,n\geq 1}, \ldots, \{A_h^{m,n}\}_{m,n\geq 1}, \tag{177}$$

are pairwise distinct, one implies that each $i$ has its corresponding $j$'s distinctly from others. Thus, $h \leq \bar{h}$. By a symmetric argument, one also has $h \geq \bar{h}$. In conclusion, one has $h = \bar{h}$. Moreover, by the above argument, for each $i$, there exists exactly one $j \in [h]$ such that $\{\bar{A}_j^{m,n}\}_{m,n\geq 1} = \{A_i^{m,n}\}_{m,n\geq 1}$. Moreover, this also implies that $B_j = B_i$.

In conclusion, there exists a permutation $\sigma \in S_h$ such that

$$\bar{A}_i^{m,n} = A_{\sigma(i)}^{m,n}, \text{ for all } m, n \geq 1, \text{ and } \bar{B}_{\sigma(i)} = B_i. \tag{178}$$

From Lemma B.12, there exists matrices $\{U_i\}_{i=1}^h \subset \mathrm{H}(d_h)$ such that

$$\bar{W}_i^Q = W_{\sigma(i)}^Q \cdot U_i^\top, \quad \bar{W}_i^K = W_{\sigma(i)}^K \cdot (U_i)^{-1}. \tag{179}$$

From the rank factorization (Piziak & Odell, 1999), there exists matrices $\{V_i\}_{i=1}^h \subset \mathrm{GL}(d_h)$ such that

$$\bar{W}_i^V = W_{\sigma(i)}^V \cdot V_i^\top, \quad \bar{W}_i^O = W_{\sigma(i)}^O \cdot (V_i)^{-1}. \tag{180}$$

This concludes the proof. $\qquad\square$

### B.6 A LEMMA CONCERNING THE ROTARY MATRIX USED IN THE PROOF OF THEOREM 3.2

Given $d = 2m$ be an even integer. Consider the RoPE matrix at position 1 as

$$R = \mathrm{diag}\big(R(\theta_1), \ldots, R(\theta_{d/2})\big) \in \mathbb{R}^{d\times d}, \text{ where } R(\theta) = \begin{bmatrix} \cos\theta & -\sin\theta \\ \sin\theta & \cos\theta \end{bmatrix}. \tag{181}$$

Denote the $n \times n$ identity matrix as $I_n$. For $i = 1, \ldots, m$, define the 2-dimensional coordinate plane

$$E_i := \mathrm{span}\{e_{2i-1}, e_{2i}\} \subset \mathbb{R}^d, \tag{182}$$

where $e_{2i-1}, e_{2i}$ are the $(2i-1)$-th and $2i$-th coordinate basis vectors. Definethe orthogonal projection matrix

$$P_i := e_{2i-1}e_{2i-1}^\top + e_{2i}e_{2i}^\top \in \mathbb{R}^{d\times d}. \tag{183}$$

In words, $P_i$ is the $d \times d$ matrix has the $i$-th $2 \times 2$ diagonal block is the $2 \times 2$ identity matrix. We also define the matrix

$$J_i := e_{2i}e_{2i-1}^\top - e_{2i-1}e_{2i}^\top \in \mathbb{R}^{d\times d}. \tag{184}$$

In words, $J_i$ is the $d \times d$ matrix has the $i$-th $2 \times 2$ diagonal block is the following $2 \times 2$ matrix

$$J := \begin{bmatrix} 0 & -1 \\ 1 & 0 \end{bmatrix}. \tag{185}$$

The matrix $R$ now can be written as

$$R = \sum_{i=1}^m \big(\cos\theta_i P_i + \sin\theta_i J_i\big). \tag{186}$$

Moreover, for $n \in \mathbb{Z}$, one has

$$R^n = \sum_{i=1}^m \big(\cos(n\theta_i)P_i + \sin(n\theta_i)J_i\big). \tag{187}$$

We have the following result.

**Lemma B.12.** *Given an integer $D \geq d$. Consider matrices $X, Z \in \mathbb{R}^{D \times d}$ and $Y, T \in \mathbb{R}^{d \times D}$. Assume that, for all non zero interger $n$,*

$$XR^nY = ZR^nT. \tag{188}$$

*If*

1. *All the angles $\theta_i \in (0, \pi)$ are pairwise distinct, and*

2. *For all $i = 1, \ldots, m$, $XP_i$ and $P_iY$ have rank 2.*

*Then, there exists an invertible matrix $U \in \mathbb{R}^{d \times d}$ of the form*

$$U = \sum_{i=1}^{m}(a_iP_i + b_iJ_i) \ \ with \ \ (a_i, b_i) \in \mathbb{R}^2 \setminus \{(0,0)\} \ for \ i = 1, \ldots, m, \tag{189}$$

*such that*

$$Z = XU \qquad and \qquad T = U^{-1}Y. \tag{190}$$

*Proof.* We structure the proof into several steps for the sake of clarity and readability.

**Step 1.**

Define

$$A_{1,i} := XP_iY \in \mathbb{R}^{D \times D}, \tag{191}$$
$$B_{1,i} := XJ_iY \in \mathbb{R}^{D \times D}, \tag{192}$$
$$A_{2,i} := ZP_iT \in \mathbb{R}^{D \times D}, \tag{193}$$
$$B_{2,i} := ZJ_iT \in \mathbb{R}^{D \times D}. \tag{194}$$

Using

$$R^n = \sum_{i=1}^{m}\left(\cos(n\theta_i)P_i + \sin(n\theta_i)J_i\right), \tag{195}$$

one has

$$
\begin{aligned}
XR^nY &= \sum_{i=1}^{m} X\left(\cos(n\theta_i)P_i + \sin(n\theta_i)J_i\right)Y \\
&= \sum_{i=1}^{m}\left(\cos(n\theta_i)XP_iY + \sin(n\theta_i)XJ_iY\right) \\
&= \sum_{i=1}^{m}\left(\cos(n\theta_i)A_{1,i} + \sin(n\theta_i)B_{1,i}\right),
\end{aligned} \tag{196}
$$

and

$$
\begin{aligned}
ZR^nT &= \sum_{i=1}^{m} Z\left(\cos(n\theta_i)P_i + \sin(n\theta_i)J_i\right)T \\
&= \sum_{i=1}^{m}\left(\cos(n\theta_i)ZP_iT + \sin(n\theta_i)ZJ_iT\right) \\
&= \sum_{i=1}^{m}\left(\cos(n\theta_i)A_{2,i} + \sin(n\theta_i)B_{2,i}\right).
\end{aligned} \tag{197}
$$

Since $XR^nY = ZR^nT$ for all $n \neq 0$, and $\theta_1, \theta_2, \ldots, \theta_m$ are pairwise distinct, one has $A_{1,i} = A_{2,i}$ and $B_{1,i} = B_{2,i}$ for all $i = 1, \ldots, m$, or

$$XP_iY = ZP_iT, \qquad and \qquad XJ_iY = ZJ_iT. \tag{198}$$

**Step 2.**

Now fix an number $i \in \{1, \ldots, m\}$. Let $X_i$ is the $D \times 2$ matrix constructed by concating the $(2i-1)$-th and $2i$-th columns of $X$, $Y_i$ be the $2 \times D$ matrix constructed by concating the $(2i-1)$-th and $2i$-th rows of $Y$. Similarly, we construct $Z_i, T_i$ for $Z, T$, respectively. By the second assumption, we have both $X_i$ and $Y_i$ have rank 2. Moreover, from

$$XP_iY = ZP_iT, \qquad \text{and} \qquad XJ_iY = ZJ_iT, \tag{199}$$

one has

$$X_iY_i = Z_iT_i, \qquad \text{and} \qquad X_iJY_i = Z_iJT_i. \tag{200}$$

Let $V_X \in \mathbb{R}^{2 \times D}$ be the left inverse matrix of $X_i$ and $V_Y \in \mathbb{R}^{D \times 2}$ be the right inverse matrix of $Y_i$,

$$V_X X_i = Y_i V_Y = I_2. \tag{201}$$

One has

$$\begin{aligned} I_2 = (V_X X_i)(Y_i V_Y) &= V_X(X_i Y_i)V_Y \\ &= V_X(Z_i T_i)V_Y = (V_X Z_i)(T_i V_Y). \end{aligned} \tag{202}$$

Let $U_i = V_X Z_i$. Then $U_i^{-1} = T_i V_Y$. Moreover, one has

$$\begin{aligned} X_i = X_i(Y_i V_Y) &= (X_i Y_i)V_Y \\ &= (Z_i T_i)V_Y = Z_i(T_i V_Y) = Z_i U_i^{-1}, \end{aligned} \tag{203}$$

so $Z_i = X_i U_i$. Similarly, one has

$$\begin{aligned} Y_i = (V_X X_i)Y_i &= V_X(X_i Y_i) \\ &= V_X(Z_i T_i) = (V_X Z_i)T_i = U_i T_i, \end{aligned} \tag{204}$$

so $T_i = U_i^{-1} Y_i$. Now, from $X_i J Y_i = Z_i J T_i$, one has

$$\begin{aligned} J = (V_X X_i)J(Y_i V_Y) &= V_X(X_i J Y_i)V_Y \\ &= V_X(Z_i J T_i)V_Y = (V_X Z_i)J(T_i V_Y) = U_i J U_i^{-1}. \end{aligned} \tag{205}$$

In other words, one has $U_i J = J U_i$. Then, there exists $(a_i, b_i) \in \mathbb{R}^2 \setminus \{(0,0)\}$ such that $U_i = a_i I_2 + b_i J$. In conclusion, one has

$$Z_i = X_i U_i, \qquad \text{and} \qquad T_i = U_i^{-1} Y_i, \tag{206}$$

where $U_i = a_i I_2 + b_i J$ with $(a_i, b_i) \in \mathbb{R}^2 \setminus \{(0,0)\}$.

**Step 3.**

Define $U = \text{diag}(U_1, \ldots, U_m)$. From the property of $U_i$'s, we have

$$U = \sum_{i=1}^{m}(a_i P_i + b_i J_i) \ \text{ with } \ (a_i, b_i) \in \mathbb{R}^2 \setminus \{(0,0)\} \ \text{ for } \ i = 1, \ldots, m, \tag{207}$$

and $Z = XU$ and $T = U^{-1}Y$. This concludes the proof. $\qquad\square$

This result will be invoked in the proof of Theorem B.11.

**Remark B.13** (On the assumptions of Lemma B.12)**.** If angles are not distinct or some equal $0$ or $\pi$, first merge blocks with equal $\theta$ and repeat the argument within each frequency class; the conclusion remains that $U$ must commute with $R$ (hence with each $J_i$) on the active subspaces. If $\text{rank}(XP_i) < 2$ or $\text{rank}(P_iY) < 2$ for some $i$, the same derivation shows $C_i$ must commute with $J_i$ on the image subspace; $C_i$ may be non-unique, but the global relation $Z = XU$, $T = U^{-1}Y$ with $U$ commuting with $R$ still describes the solution set restricted to the active coordinates.

## C    ALGORITHM DESCRIPTION

---

**Algorithm 1** Teleportation Training with Sampling Minimal Perturbations.

---

**input** Loss function $\mathcal{L}(w)$, optimizer $\varphi$, number of optimization steps $T$, initialization $\theta_0 \in \Theta$, teleportation steps $K$, perturbation range $\alpha > 0$, number of samples $M$.
1: **for** $t \leftarrow 0$ to $T - 1$ **do**
2:     **if** $t \in K$ **then**
3:         Sample a set of perturbations $B = \{g_i \in B_G(\alpha)\}_{i=1}^M$
4:         $S \leftarrow \{g \in B \mid \|\nabla\mathcal{L}(g\theta_t)\|_2 > \|\nabla\mathcal{L}(\theta_t)\|_2\}$
5:         **if** $|S| > M/2$ **then**
6:             Find the best perturbation: $g^* \leftarrow \arg\max_{g \in B} \|\nabla\mathcal{L}(g\theta_t)\|_2$
7:             $\theta_t \leftarrow g^*\theta_t$
8:         **end if**
9:     **end if**
10:     $\theta_{t+1} \leftarrow \varphi(\theta_t)$
11: **end for**
**output** $\theta_T$

---

## D  OPTIMIZER CONSIDERATIONS FOR TELEPORTATION

The Adam optimizer (Adam et al., 2014) maintains exponential moving averages of the gradient and its elementwise square. Given the stochastic gradient $g_t = \nabla_\theta\mathcal{L}(\theta_t)$ at iteration $t$, the moment estimates are defined as:

$$m_t = \beta_1 m_{t-1} + (1 - \beta_1)g_t, \tag{208}$$

$$v_t = \beta_2 v_{t-1} + (1 - \beta_2)g_t^2, \tag{209}$$

where $\beta_1, \beta_2 \in [0, 1)$ denote exponential decay rates for the first and second moments, respectively. To correct for initialization bias, the estimates are normalized as:

$$\widehat{m}_t = \frac{m_t}{1 - \beta_1^t}, \qquad \widehat{v}_t = \frac{v_t}{1 - \beta_2^t}. \tag{210}$$

The parameter update rule is then:

$$\theta_{t+1} = \theta_t - \eta\frac{\widehat{m}_t}{\sqrt{\widehat{v}_t} + \epsilon}, \tag{211}$$

with learning rate $\eta > 0$ and numerical stabilizer $\epsilon > 0$. Since both $m_t$ and $v_t$ scale proportionally with $g_t$, the effective update $\widehat{m}_t/\sqrt{\widehat{v}_t}$ normalizes gradient magnitude. Consequently, increases in $|g_t|$—such as those induced by teleportation—do not translate into proportionally larger parameter updates. This adaptivity dampens the sensitivity of Adam to gradient-norm amplification. By contrast, stochastic gradient descent (SGD) applies the update $\theta_{t+1} = \theta_t - \eta g_t$, where the step size scales linearly with $\|g_t\|$, thereby preserving the full effect of teleportation-induced gradients.

Beyond this difference, the broader literature has reported several shortcomings of Adam relative to SGD. In particular, Adam may fail to guarantee convergence in certain regimes (Reddi et al., 2019), and often yields inferior generalization despite faster initial progress (Wilson et al., 2017). These limitations have been linked to over-reliance on momentum dynamics and misalignment between adaptive updates and descent directions (Gitman et al., 2019). In contrast, SGD has been shown to encourage flatter minima and superior generalization properties in deep learning models (Zhou et al., 2020; Chen et al., 2018).

Taken together, these considerations suggest that SGD is generally more favorable than Adam in the context of teleportation. Since teleportation deliberately amplifies gradient signals, Adam's adaptive normalization tends to attenuate its effect, whereas SGD preserves the proportional update and better leverages the intended perturbations. Therefore, the majority of experiments in this work employ SGD as the base optimizer.

## E  EXPERIMENTAL DETAILS AND HYPERPARAMETERS

Our experiments are designed to evaluate the effect of teleportation across both vision and language modeling benchmarks. For vision tasks, the evaluation covers MNIST, CIFAR-10, and ImageNet-1K, while for language modeling the benchmark is WikiText-103. SGD with a cosine learning-rate

schedule is employed in all experiments. The study focuses exclusively on teleportation within attention layers, which are modified in all Transformer layers, ReLU is used as the activation functionm and teleportation is not applied to FFN components (i.e MLP blocks). In our experimental setup, learnable APE is adopted for vision task, while sinusoidal APE is applied to the WikiText-103.

For robustness, each configuration on MNIST and CIFAR-10 is repeated for five independent runs, while WikiText-103 and ImageNet-1K experiments are repeated three times per configuration.

Table 3: GPU Memory Allocated (Gb) on MNIST and CIFAR-10 (smaller is better).

| Datasets | PE | No Teleport | Teleport | Zhao's Teleport |
|----------|-----|-------------|----------|-----------------|
| MNIST | APE | 1.14 | 1.16 | **2.36** |
|  | RoPE | 1.17 | 1.18 | **2.23** |
| CIFAR-10 | APE | 2.03 | 2.07 | **5.30** |
|  | RoPE | 2.06 | 2.09 | **5.02** |

**MNIST.** The experiments are conducted using a variant of ViT-Tiny with 6 transformer layers, hidden size of 128, MLP hidden dimension of 512, 4 self-attention heads, and attention dropout rates set to 0.0. Models are trained for 20 epochs with a batch size of 128, an initial learning rate of 0.015, momentum of 0.9, and weight decay of 1e-4. Teleportation is applied once at epoch 1 with a radius of 0.65 ($\alpha = 0.65$), covering the first 4 consecutive steps ($|K| = 4$). At each teleportation step, 16 matrices are sampled ($M = 16$).

**CIFAR-10.** The experiments are conducted using a variant of ViT-Tiny with 6 transformer layers, hidden size of 192, MLP hidden dimension of 768, 3 self-attention heads, and hidden and attention dropout rates set to 0.0. Training is performed for 50 epochs with a batch size of 256, an initial learning rate of 0.005, momentum of 0.9, and weight decay of 1e-5. Teleportation is applied once at epoch 1 with a radius of 0.65 ($\alpha = 0.65$), covering the first 4 consecutive steps ($|K| = 4$), with 16 matrices sampled per step ($M = 16$).

**ImageNet-1K.** The experiments are conducted using the ViT-Tiny-Patch16-224 architecture, configured with 12 Transformer layers, a hidden size of 192, MLP hidden dimension of 768, and 3 self-attention heads. The encoder employs a patch size of 16, ReLU is used as the activation function, with both attention and hidden dropout rates set to 0.0, a initial learning rate of 0.05, batch size of 256, warmup learning rate of 1e-7, and a minimum learning rate of 1e-5. Teleportation is applied starting from epoch 2 with a radius of 0.2. At each teleportation step, 8 matrices are sampled ($M = 8$), and a total of 32 teleportation steps are executed ($|K| = 32$), divided into two sessions of 16 consecutive steps.

**WikiText-103.** The experiments are conducted using a Transformer-XL architecture with 16 layers, model dimension 128, inner dimension 2048, 8 attention heads with head dimension 16. The target length and evaluation length are set to 256, and no memory is carried across segments (mem_len=0). The dropout rate is 0.1, and attention dropout is set to 0.0. Training is performed with using an initial learning rate of 0.75, warmup over 2000 steps, and with a batch size of 96. Teleportation is applied only to the attention layers, beginning at step 500 and continuing through step 515 ($|K| = 16$). At each teleportation step, 8 matrices are sampled ($M = 8$) with a scaling radius 0.2 ($\alpha = 0.2$).

**Zhao et al. (2023) algorithm.** We adopt the same model architectures and optimization hyperparameters as described above for MNIST and CIFAR-10. The teleportation configuration is kept at the default settings across both datasets, specifically: the teleportation learning rate of 1e-4, the teleportation step of 10 (referring to the number of gradient ascent iterations for optimizing $g$, which differs from our definition of teleportation steps), the teleportation epoch of 3, and the total of 8 steps being teleported.

All experiments were carried out on a single NVIDIA H100 GPU with 80GB of memory, while the maximum VRAM actually used did not exceed 26GB. Training on MNIST and CIFAR finishes

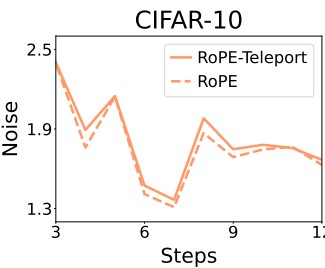 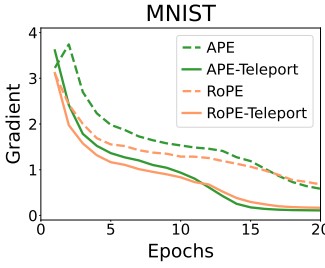 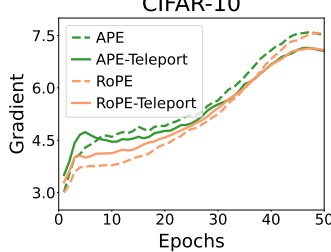

(a) Gradient noise increased after applying teleportation in step 3.

(b) $\ell_2$ gradient norms on MNIST and CIFAR-10, where teleportation results in smaller values relative to the non-teleportation baseline.

Figure 3: Demonstrates the generalization of teleportation via gradient noise and $\ell_2$ gradient norm.

within 7 minutes, whereas large-scale runs take considerably longer–up to 90 hours for ImageNet-1K and 33 hours for WikiText-103.

**Comparison of GPU allocation between our and zhao algorithms.** Zhao's algorithm consumes twice as much GPU memory but does not bring any significant effect on validation accuracy or convergence time (Table 1, Table 3).

**Sharpness on MNIST and CIFAR-10 after teleportation.**

Table 4: Sharpness on MNIST and CIFAR-10 (smaller is better).

| Datasets | PE | No Teleport | Teleport |
|---|---|---|---|
| MNIST | APE | $2844.37 \pm 512.19$ | $1168.78 \pm 298.98$ |
| | RoPE | $98.47 \pm 25.31$ | $95.23 \pm 20.34$ |
| CIFAR-10 | APE | $1054.52 \pm 78.27$ | $958.78 \pm 63.30$ |
| | RoPE | $484.67 \pm 56.55$ | $434.06 \pm 40.04$ |

**Gradient noise and $\ell_2$ gradient norm after teleportation on CIFAR-10.**

## F   TELEPORTATION FOR ADAM

We further evaluate the effect of teleportation when training with the AdamW (Loshchilov & Hutter, 2017) optimizer. The network architectures and training hyperparameters follow details provided in Appendix E. For MNIST, training is conducted with a batch size of 128 for 20 epochs, an initial learning rate of 2.5e-4, and weight decay 1e-5. For CIFAR-10, we use a batch size of 256, 50 training epochs, the same initial learning rate 2.5e-4, and weight decay 1e-5.

Teleportation is applied exclusively to attention layers, with no modification to MLP components, and the number of samples is fixed at $M = 16$ per teleportation step. For MNIST (both APE and RoPE positional embeddings), teleportation is performed at epochs 1–3, with 8 consecutive steps at the beginning of each epoch ($|K| = 24$) and radius 0.1 ($\alpha = 0.1$). For CIFAR-10 with learnable embeddings, the same schedule is applied but with radius 0.2. For CIFAR-10 with RoPE, teleportation is performed only at epoch 1, consisting of 16 consecutive steps ($|K| = 16$) with radius 0.2 ($\alpha = 0.2$).

Overall, Table 5 shows that teleportation with AdamW yields only marginal gains in validation accuracy over the non-teleportation baseline. Improvements in training time are inconsistent and considerably smaller than those observed with SGD, suggesting that teleportation is less effective when combined with adaptive optimizers such as Adam and AdamW.

Table 5: Results of teleportation with AdamW on MNIST and CIFAR-10. Reported are mean and standard deviation over five independent runs.

| Dataset | PE | Teleport | Val Acc (%) ↑ | Speedup (%) ↑ | Time/epoch ↓ |
|---------|-----|----------|----------------|-----------------|----------------|
| MNIST | APE | No | $98.81 \pm 0.07$ | - | $7.83 \pm 0.82$ (s) |
| | | Yes | $98.83 \pm 0.08$ | $20.83 \pm 4.17$ | $8.37 \pm 0.68$ (s) |
| | RoPE | No | $99.06 \pm 0.00$ | - | $9.17 \pm 1.04$ (s) |
| | | Yes | $99.08 \pm 0.05$ | $6.54 \pm 4.58$ | $10.25 \pm 0.75$ (s) |
| CIFAR-10 | APE | No | $78.18 \pm 0.28$ | - | $7.11 \pm 0.60$ (s) |
| | | Yes | $78.36 \pm 0.27$ | $13.13 \pm 11.46$ | $7.23 \pm 0.63$ (s) |
| | RoPE | No | $80.98 \pm 0.25$ | - | $6.57 \pm 0.35$ (s) |
| | | Yes | $81.69 \pm 0.53$ | $11.40 \pm 10.66$ | $6.72 \pm 0.42$ (s) |

## G  TELEPORTATION CONFIGURATION RECOMMENDATIONS

**Hyperparameter trade-off.** The effectiveness of teleportation is governed by multiple interacting factors, including the radius $\alpha$ and the choice of teleportation steps $K$. Both need to be tuned with care depending on dataset size and model architecture. When teleportation is applied to later training stages, a smaller radius is preferable since gradients are already relatively stable at this point, and large perturbations may cause undesirable fluctuations. Conversely, a larger radius $\alpha$ is typically applied in earlier stages and can be stabilized with fewer consecutive teleportation steps.

**Recommended configuration.**

- **Radius $\alpha$:** Choose $\alpha \in [0.2, 0.6]$. Larger radius ($\geq 0.5$) work best with 4–6 consecutive steps; medium radius (0.3–0.5) with 6–10 steps; and smaller radius ($\leq 0.3$) with 10–16 steps. These recommendations are derived from our empirical observation that the cumulative ratio of gradient norms (after teleportation/before teleportation) across consecutive steps should remain below 1.05 for small datasets (e.g., CIFAR-10, MNIST) and close to 1.00 for large datasets (e.g., ImageNet-1K, WikiText-103) to avoid gradient explosion.

- **Total teleportation steps $|K|$:** For smaller datasets, $|K|$ should be around 2–4% of the number of training iterations per epoch. For larger datasets, $|K| \approx 0.5\%$ is sufficient.

- **Consecutive steps:** Should not exceed 16, and generally should not be fewer than 4 to have noticeable effect. Total teleportation steps should not exceed twice the number of consecutive steps (i.e., only 1–2 consecutive teleportation phases per run).

- **Teleportation position:** Empirical evidence suggests that teleportation is most effective when scheduled within the first 5 epochs. In the absence of learning rate warm-up, it should be applied during the earliest epochs, at the stage where the loss is decreasing most rapidly and before convergence stabilizes. With warm-up, teleportation is better placed in the middle of the warm-up phase.

- **Sampling $M$:** Use 8–16 samples. Fewer than 8 leads to instability, while more than 16 brings little additional benefit.

The above recommendations are intended as a practical guideline for deploying teleportation in training pipelines. They have been validated across both vision and NLP benchmarks, and strike a balance between stability and efficiency. While adjustments may be explored for further empirical gains, substantial deviations from these ranges tend to introduce instability and are therefore not advised unless carefully evaluated.

## H  TELEPORTATION INDEX

Table 6: Effect of teleportation index on WikiText-103 performance.

| Teleport index | Val PPL ↓ | Test PPL ↓ | Teleport index | Val PPL ↓ | Test PPL ↓ | Teleport index | Val PPL ↓ | Test PPL ↓ |
|---|---|---|---|---|---|---|---|---|
| 0–500 | 35.13 | 36.10 | 1500–2000 | 34.87 | 35.98 | 0–2000 | 34.69 | 35.87 |
| 500–1000 | **34.39** | **35.45** | 2000–2500 | 35.94 | 36.83 | 0–4000 | 34.71 | 35.86 |
| 1000–1500 | 35.23 | 36.21 | 2500–3000 | 35.08 | 36.09 | 2000-4000 | 35.12 | 36.17 |

