# OpenReview forum: "Accelerating Transformer Training: Architectural Symmetry, Positional Encoding, and Teleportation"
_ICLR.cc/2026/Conference — ICLR 2026 Conference Withdrawn Submission_

### Official Review · Reviewer_GkSi · 2025-10-27

**Soundness:** 2
**Presentation:** 3
**Contribution:** 2
**Rating:** 4
**Confidence:** 4

**Summary:**

Based on a systematic study of multi-head attention and positional embeddings, this paper successfully extends the teleportation technique to Transformer architectures. Moreover, to avoid using expensive Hessian-based methods, the authors propose finding an optimal$g \in G$ through small perturbations. Experimental results demonstrate its effectiveness on Transformer models for small vision and NLP tasks.

**Strengths:**

- The paper is well structured and easy to follow.
- A detailed analysis of teleportation in Transformer architectures is provided.

**Weaknesses:**

- The experiments are conducted only on very small datasets. Since Transformer architectures are typically applied to large-scale datasets, the current experimental settings are not very convincing.
- What is the definition of "speedup" in Table 1? The “Time/epoch” for teleport-based training is higher than that of the baseline without teleportation, which seems inconsistent with the claimed “Speedup.”
- Although the authors discuss and claim that the cost of perturbation sampling remains below 3% in their small-scale datasets and model settings, such overhead can be significant in large-scale Transformer pre-training under distributed settings.
- Why is the speedup for the AdamW optimizer limited compared with SGD? As most large-scale Transformer models are trained with AdamW (or Muon), this limitation reduces the practical value of the proposed method.

**Questions:**

See weaknesses

---

### Official Review · Reviewer_PThV · 2025-10-27

**Soundness:** 3
**Presentation:** 3
**Contribution:** 3
**Rating:** 4
**Confidence:** 3

**Summary:**

The presented paper provides an explicit construction of the "maximal symmetry group" of self-attention with ROPE as its positional encoding. It also proposes a new teleportation method that introduces minimal perturbation and is easy to compute.

**Strengths:**

- The explicit construction of the maximal symmetry group of MHA with ROPE seems pretty novel and strong.
- The presentation is clear, and the paper is easy to follow.
- I do appreciate the study of the maximal symmetry group, because it characterizes the boundary of symmetry-based methods. I believe this concept and the results proved in this paper can be beneficial to the community.

**Weaknesses:**

Firstly, the whole section 4 looks pretty strange to me, for the following reasons:
- I don't see a connection between Section 4 and previous sections. It seems like they are discussing totally different topics. In my understanding, the key idea of Section 2 and 3 is to derive a formula of the maximal symmetry group of MHA with ROPE, while in Section 4 the authors suddenly turn to discuss how to perform teleportation effectively. I don't see a clear logical connection between these two topics.
- Also, I don't understand the idea behind "minimal perturbation". The entire concept of symmetry-based teleportation is to explore the weight space without changing the functionality, isn't it? If we keep the perturbation small, the small perturbation itself is enough to keep the functionality not changing too much, then what's the point of using symmetric transformations as teleportation?
- In line 320, the authors claim: "each teleportation step is now performed within this ball ...". However, later in the actual implementation (line 330) "we sample near the identity by constructing a diagonal matrix as follows...", but the diagonal matrices do not form a ball in GL(n). Also, if the teleportation is limited to a zero-measure set of the symmetry group (the set of all diagonal matrices), what's the point of exploring the maximal symmetry group?

Other weaknesses & questions:
- To me the most novel and important result of this paper is perhaps Theorem 3.2. However, the main paper does not mention how to prove this theorem. I hope the authors can at least write a proof sketch and the ideas behind it.
- There is no need at all to discuss the absolute positional encoding. Why not remove the whole Section 3.1 and save the sapce to write a proof sketch of Theorem 3.2?
- The title and abstract is somewhat misleading. This paper actually only focuses on the symmetry in the self-attention module (which is reasonable, since the symmetry in FFN is the same as those in MLP and has been extensively studied). The current title and abstract is too broad. It's better to limit the scope to multi-head attention. The current title sounds like a survey paper.
- Although the experiment results look good, they are limited to small datasets. and the ablation study is not sufficient enough. For example, I think this is the key question the ablation study should answer: to outperform the baseline, which part is the most critical? the "small perturbation", the sampling of $g$, or the stability truncation?

Minor issues and questions:
- What is "proper real algebraic variety"?
- What does $\nabla \mathcal L|_{g\theta}$ mean in eq.(13)? Do you mean the gradient at $g \theta$?
- In eq. (13), why minimize the operator norm of the gradient?
- Definition 2.2 is not needed. Actually, I think the definition of "maximal" or "small exception" is still unclear to me. Is it possible to rewrite is with the language of, say, measures?

**Questions:**

See Weaknesses.

---

### Official Review · Reviewer_V6vP · 2025-10-31

**Soundness:** 3
**Presentation:** 3
**Contribution:** 2
**Rating:** 6
**Confidence:** 3

**Summary:**

The paper discusses the symmetry group that occurs in the weight space of the self-attention component, with a focus on attention with rotary-positional encodings. It extends the ideas from previous work [1] by including the positional encodings. Then it proceeds to propose an algorithm for training with so called teleportation where in some steps the parameters are perturbed to values that result in a functionally-equivalent model but with larger gradient norms. Finally, the paper demonstrates positive effects of the proposed methods on the convergence speed of SGD and the generalization of the obtained model.

[1] Hoang V. Tran, Thieu Vo, An Nguyen The, Tho Tran Huu, Minh-Khoi Nguyen-Nhat, Thanh Tran, Duy-Tung Pham, and Tan Minh Nguyen. Equivariant neural functional networks for transformers. In The Thirteenth International Conference on Learning Representations, ICLR, 2025.

**Strengths:**

1. The presentation of the paper is very good. The problem and prerequisites are well outlined and the main result clearly presented. I especially appreciate the shortened versions of the proofs in the appendix before stating the whole proof.
2. For my understanding, finding a symmetry group that takes the embedding into account is novel.
3. The proposed method seems to improve the convergence speed of Transformer models in the studied settings.
4. The paper provides an ablation study of the proposed method together with recommendations on how to set the hyperparameters.

**Weaknesses:**

1. The paper does not explain how the algorithm accounts for the symmetry group H from equation 9 in case of models with RoPE embedding. The procedure described in lines 329-335 explains only how to sample from GL(n).
2. The optimizer of choice while training Transformers is usually Adam(W) and not SGD, yet the proposed method mainly improves the performance of SGD with only a tiny improvement for AdamW. The method applied to AdamW is also not tested on a language task only on MNIST and CIFAR-10 (table 5).

**Questions:**

1. What exactly is meant by sharpness in Table 4? Is it some Hessian-based measure or something else?
2. When used on an architecture with RoPE, how does the method sample the set of perturbations for the query and key parameters so that they are part of H from equation 9?
3. How does the method perform when the model is trained with AdamW on a language task?
4. Does the teleportation + SGD outperform training with AdamW? I would be curious to see a figure similar to Figure 2 comparing these two settings.
5. When adding teleportation to an optimizer, is it necessary to retune optimizer hyperparameters? I would ideally like to see some experimental evidence to whether that is the case or not.
6. In line 429 the paper states that the teleportation can be applied also to the FFN part of the network. What is the symmetry group used in this case? Does it take activations into account?

**Details Of Ethics Concerns:**

I would like to flag this submission as potentially violating the submission policy. Submission 5096 presents the same theoretical result on the characterisation of the symmetry of attention with widely used PE, and I have reasons to believe (see below) that both papers share authors. The papers use the theoretical result in different application contexts (LMC and efficient Transformer training) but they both claim that the theoretical result and its discussion are one of their main contributions.

The reason I believe that the papers share authors and claim the same contributions are the following parts of the papers:
* In “Contribution” section of submission 5096 in lines 110-117 we read:
>>2. In Section 3, we analyze how positional encodings alter the internal structure of attention. We focus primarily on the most widely used encodings, Absolute PE and Relative PE. In particular, we study sinusoidal PE as a representative of APE and rotary PE as a representative of RPE, and show why results from the vanilla case do not extend directly to these settings.
>> 3. In Section 4, we present the main result of the paper, which characterizes the full symmetry of attention with widely used positional encodings. This characterization underlies the matching algorithm for Multihead Attention described in Section 5.

Meanwhile, in “Contribution” section of submission 5098 in lines 98-102 we read:
>> 2. In Section 3, we analyze how positional encodings alter the internal structure of attention. We focus primarily on the most widely used encodings, Absolute PE and Relative PE. In particular, we study sinusoidal PE as a representative of APE and rotary PE as a representative of RPE, and show why results from the vanilla case do not extend directly to these settings. We then present our finding that fully characterizes the symmetry of attention with widely used PE.
* The symmetry group under RoPE both papers introduce is the same, just with slightly different notation, see lines 219-247 in 5096 and lines 249-278 in 5098.
* The main theoretical result regarding multihead attention with RoPE is the same in both papers with just slightly altered exposition, see lines 308-320 in 5096 (Theorem 4.2) and lines 279-289 in 5098 (Theorem 3.2).
* Both papers derive their theorem as a consequence of a more general result. Submission 5096 states it lines 289-292 (Theorem 4.1) while submission 5098 in lines 895-903 in the appendix (Theorem B.1). The proofs of both theorems also look strikingly similar (see lines 1499-1527 in submission 5096 and lines 905-933 in submission 5098).

---

### Official Review · Reviewer_BQtQ · 2025-11-03

**Soundness:** 3
**Presentation:** 3
**Contribution:** 3
**Rating:** 4
**Confidence:** 2

**Summary:**

The paper first analyzes how multihead attention symmetry is affected by sinusoidal and rotary positional encodings. Then this paper introduces a teleportation framework for transformer-based on these insights, and demonstrates its effectiveness across various settings, highlighting the relationship between positional encoding, symmetry, and optimization.

**Strengths:**

1. This paper provides a bridge for the practical application of the teleportation framework in deep learning training.
2. This paper is well-organized and easy to read.

**Weaknesses:**

1. In Table 1, since CIFAR and MNIST datasets are not large enough to demonstrate the generalization, further experiments on ImageNet1K comparing with other teleportation methods are needed to demonstrate the effectiveness of the proposed framework.

**Questions:**

Please see the weakness.

---

### Author Response · Authors · 2025-11-26
**Withdrawal Notice**

Dear Reviewers and AC,

We sincerely thank all Reviewers for their thoughtful feedback, which has greatly helped us identify areas for improvement in our manuscript. We also thank the AC for their effort in ensuring a smooth and fair review process.

After careful consideration, we believe that properly addressing the concern raised by Reviewer V6vP regarding overlapping content would require substantial revision of the manuscript. This is our oversight, and we take full responsibility for it. Therefore, we intend to withdraw this submission at the conclusion of the review process and revise it for a future venue.

Once again, we thank the Reviewers for their time, effort, and constructive comments.

Best regards,

The Authors

---

### Note · Authors · 2026-01-08

I have read and agree with the venue's withdrawal policy on behalf of myself and my co-authors.